# Visual Thoughts: A Unified Perspective of Understanding Multimodal Chain-of-Thought

**Zihui Cheng**[1,4*]   **Qiguang Chen**[2*]   **Xiao Xu**[2]   **Jiaqi Wang**[5]   **Weiyun Wang**[6]   **Hao Fei**[7]
**Yidong Wang**[8]   **Alex Jinpeng Wang**[1]   **Zhi Chen**[9]   **Wanxiang Che**[2]   **Libo Qin**[1,3,4†]

[1] School of Computer Science and Engineering, Central South University
[2] Research Center for Social Computing and Interactive Robotics, Harbin Institute of Technology
[3] Institute of Computing and Intelligence, Harbin Institute of Technology, Shenzhen
[4] Text Computing and Cognitive Intelligence Ministry of Education Engineering Research Center,
Guizhou University    [5] Chinese University of Hong Kong    [6] Shanghai AI Laboratory
[7] National University of Singapore    [8] Peking University    [9] ByteDance Seed (China)
`czh_up@csu.edu.cn, qgchen@ir.hit.edu.cn, qinlibo@hit.edu.cn`

## Abstract

Large Vision-Language Models (LVLMs) have achieved significant success in multimodal tasks, with multimodal chain-of-thought (MCoT) further enhancing performance and interpretability. Recent MCoT methods fall into two categories: (i) Textual-MCoT (T-MCoT), which takes multimodal input and produces textual output; and (ii) Interleaved-MCoT (I-MCoT), which generates interleaved image-text outputs. Despite advances in both approaches, the mechanisms driving these improvements are not fully understood. To fill this gap, we first reveal that MCoT boosts LVLMs by incorporating *visual thoughts*, which convey image information to the reasoning process regardless of the MCoT format, depending only on clarity and conciseness of expression. Furthermore, to explore visual thoughts systematically, we define four distinct forms of visual thought expressions and analyze them comprehensively. Our findings demonstrate that these forms differ in clarity and conciseness, yielding varying levels of MCoT improvement. Additionally, we explore the internal nature of visual thoughts, finding that visual thoughts serve as intermediaries between the input image and reasoning to deeper transformer layers, enabling more advanced visual information transmission. We hope that the visual thoughts can inspire further breakthroughs for future MCoT research.

## 1   Introduction

Recently, Large Vision-Language Models (LVLMs) have made remarkable advancements in tackling various multimodal tasks [57, 22, 52]. Inspired by the success of Chain-of-Thought (CoT) reasoning [45, 17, 3], LVLMs have integrated Multi-modal CoT (MCoT) reasoning, enabling step-by-step generation of reasoning paths in multimodal contexts. This advancement has enhanced their reasoning abilities, facilitating more sophisticated interactions with multimodal inputs [54, 4]. Specifically, current MCoT techniques are generally divided into two paradigms: **(1) Textual-MCoT (T-MCoT)**: This approach adheres to the traditional CoT framework, generating text-based rationales from multimodal inputs (see Figure 1 (a)). For example, some methods require LVLMs to interpret and describe visual elements before producing an answer [55, 51], while others enhance reasoning by integrating json-format scene graphs derived from images [30, 31]. **(2) Interleaved-MCoT (I-MCoT)**: A more recent approach, such as o3-mini [33] and Visual Sketchpad [15], generates interleaved image-text

---

[*]Equal Contribution
[†]Corresponding Author

39th Conference on Neural Information Processing Systems (NeurIPS 2025).

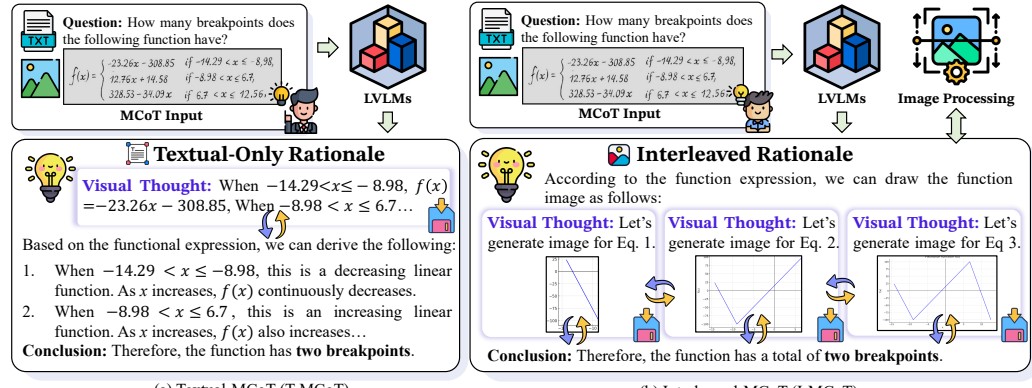

(a) Textual-MCoT (T-MCoT)        (b) Interleaved-MCoT (I-MCoT)

Figure 1: Comparison between (a) Textual MCoT (T-MCoT) with purely textual rationale, and (b) Interleaved MCoT (I-MCoT) with the image-text interleaved rationale. VT: visual thoughts.

rationales (see Figure 1 (b)). This method employs external tools, like code interpreters or specialized visual models, to modify images for reasoning [28, 15, 56], while others use image generation models to create new images, enhancing the reasoning process [27, 19].

However, the debate between these two paradigms remains unresolved. Some researchers contend that the interleaved rationale in I-MCoT better reflects human cognitive processing of multimodal inputs, potentially offering advantages over T-MCoT [15, 19]. In contrast, other studies suggest that in mathematical contexts, text-only rationales may yield superior performance [56, 7]. This disagreement highlights a fundamental gap in understanding the mechanisms behind different MCoT approaches. In addition, in the literature, there is a lack of a unified framework to explain MCoT's effectiveness, identify optimal MCoT paradigms, or derive generalizable insights across tasks. Motivated by this, in this work, we aim to investigate the following research question: ***Is there a unified explanation for that different MCoT paradigms enhance LVLMs in distinct ways?***

To this end, we reveal that MCoT unifiedly improves LVLMs by integrating **visual thoughts** [18] into the rationale generation. Visual thoughts are intermediate, logic-driven cross-modal representations that facilitate and accelerate multimodal reasoning within a unified perspective. By caching distilled visual information, they bridge raw pixels and linguistic rationales, enabling fast, context-aware access without reprocessing the image. Conceptually, as shown in Figure 2, image features and visual thoughts function like external memory and cache: raw images require slow, resource-intensive retrieval, while visual thoughts retain key content for rapid access, improving reasoning efficiency and reducing computational cost. Moreover, the effectiveness and efficiency of visual thought transmission can further explain performance differences across MCoT paradigms.

To verify this, during experiments, we first verify the effectiveness of visual thoughts in improving MCoT performance. Then, to extensively explore how visual thoughts work in different expressions, we categorize four major strategies: *Natural Language*, *Structured Language*, *Edited Image*, and *Generative Image* visual thoughts. Moreover, we also investigate the role of internal attention mechanisms and information flow within LVLMs to analyze the rationale behind visual thoughts. Our findings reveal the following: (1) *Removing visual thoughts and forcing reasoning solely from the original image can impair performance, even more than reasoning directly from the query.* (2) *Different expressions of visual thoughts are more effective in certain scenarios, depending on their expression clarity and efficiency.* (3) *Visual thoughts not only carry visual information but also serve as primary intermediaries, connecting the input image to deeper transformer layers and enabling more advanced cognitive processing in LVLMs.*

Our main contributions can be summarized as follows:

- We present the first comprehensive exploration of the underlying mechanisms driving visual thoughts in MCoT reasoning. This unified perspective provides novel insights into how visual cognition unfolds during decision-making processes.
- We introduce four distinct strategies for systematically exploring visual thoughts, demonstrating the advantages and disadvantages of different strategies.
- Our analysis further delves deeply into the nature of visual thoughts, revealing critical insights, such as the way in which visual information is integrated into the reasoning path, which enables

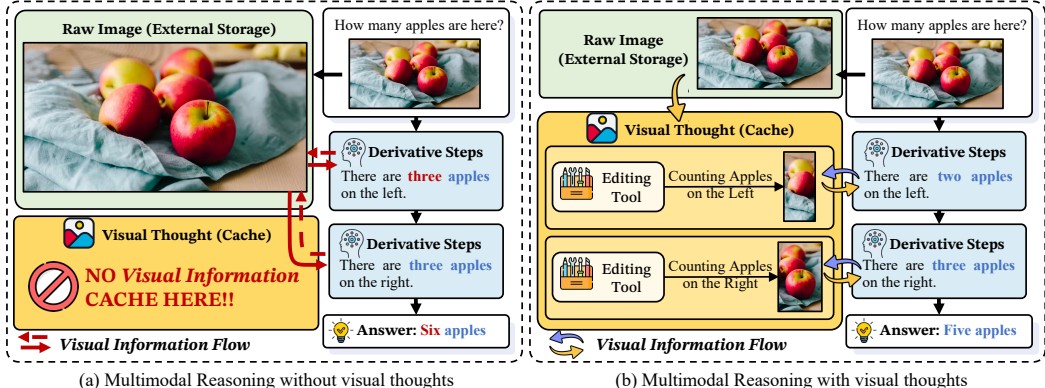

Figure 2: Comparison of multimodal reasoning from a computer-system perspective: (a) visual thoughts as an internal visual cache versus (b) direct access to raw images as external storage.

conveying more visual information into the deeper transformer layers of LVLMs. We hope these findings will inspire more breakthroughs for future research in this area.

## 2 Visual Thoughts

### 2.1 Definition of Visual Thoughts

To explain how MCoT improves performance, we contend that its primary benefit is the incorporation of **visual thoughts**, intermediate reasoning steps that explicitly convey visual information and enable LVLMs to perform deeper visual reasoning. As shown in Figure 2, LVLMs treat the raw image as external memory, forcing the model to iteratively reprocess the entire visual input at each step so as to cap the depth of reasoning. In contrast, visual thoughts extract only the instruction-relevant regions (e.g., the left and right apples) and store them as a cache. Subsequent reasoning steps query this cache rather than the full image, reducing computational overhead and enabling deeper, multi-step MCoT.

Formally, a visual thought ($\mathcal{R}^{\mathcal{E}_{VT}}$) is a reasoning step that conveys information from the visual input $\mathcal{V}_I$ and all previous steps $\mathbf{R}_{<i} = \{\mathcal{R}_1, \mathcal{R}_2, \ldots, \mathcal{R}_{i-1}\}$. These steps are driven by the task question $\mathcal{Q}_T$ and by explicit instructions $\mathcal{I}_{\mathcal{E}}$ requesting the MCoT expression $\mathcal{E}_{VT}$. The model then generates the next reasoning step $\mathcal{R}_i$ as follows:

$$\mathcal{R}_i = \begin{cases} \underset{\mathcal{R}^{\mathcal{E}_{VT}}}{\operatorname{argmax}} \, \dot{\pi}\left(\mathbf{R}_{<i}, \mathcal{V}_I \rightarrowtail \mathcal{R}^{\mathcal{E}_{VT}} | \mathcal{V}_I, \mathcal{Q}_T, \mathcal{I}_{\mathcal{E}_{VT}}, \mathbf{R}_{<i}\right), & \text{if } \dot{\pi} \geq \pi, \\ \underset{\mathcal{R}^D}{\operatorname{argmax}} \, \pi(\mathbf{R}_{<i}, \mathcal{R}^{\mathcal{E}_{VT}} \rightarrowtail \mathcal{R}^D | \mathcal{V}_I, \mathcal{Q}_T, \mathcal{I}_{\mathcal{E}_{VT}}, \mathbf{R}_{<i}), & \text{if } \dot{\pi} < \pi, \end{cases} \quad (1)$$

where $\mathcal{R}^D$ represents the derivative reasoning steps [58] following $\mathcal{R}_i$ and grasp visual information from $\mathcal{R}^{\mathcal{E}_{VT}}$. The function $\dot{\pi}(\cdot)$ denotes the probability of generating visual thoughts, and $\pi(\cdot)$ represents the probability of generating derivative reasoning steps. The symbol $x \rightarrowtail y$ indicates the reasoning step $y$ with the flow of reasoning information from $x$ to $y$.

### 2.2 Categories of Visual Thoughts

Visual thoughts can be expressed in different modalities depending on the MCoT variant: (1) In T-MCoT, which generates text-based rationales, they appear as *textual expressions*; (2) In I-MCoT, which produces cross-modal rationales, they manifest as *visual expressions*.

#### 2.2.1 Textual Multimodal Chain-of Thought (T-MCoT)

We first define visual thoughts within T-MCoT, where the model generates visual thoughts as textual tokens. As shown in Figure 1 (a), the traditional T-MCoT generates text-only output from multimodal inputs, representing visual thoughts as $\mathcal{E}_{VT} = \mathcal{E}_{text}$. Formally, the visual thought can be expressed as:

$$\mathcal{R}_i = \underset{\mathcal{R}^{\mathcal{E}_{text}}}{\operatorname{argmax}} \, \pi_{text}(\mathbf{R}_{<i}, \mathcal{V}_I \rightarrowtail \mathcal{R}^{\mathcal{E}_{text}} | \mathcal{V}_I, \mathcal{Q}, \mathcal{I}_{\mathcal{E}_{text}}, \mathbf{R}_{<i}), \quad (2)$$

where $\pi_{text}(\cdot)$ denotes the probability of generating the rationale $\mathcal{R}^{\mathcal{E}_{text}}$ from the textual tokens.

*Expression 1:* **Natural Language ($N$-LANG)** facilitates effective visual information transfer by natural language expression, such as describing images based on question, enhancing vision-language

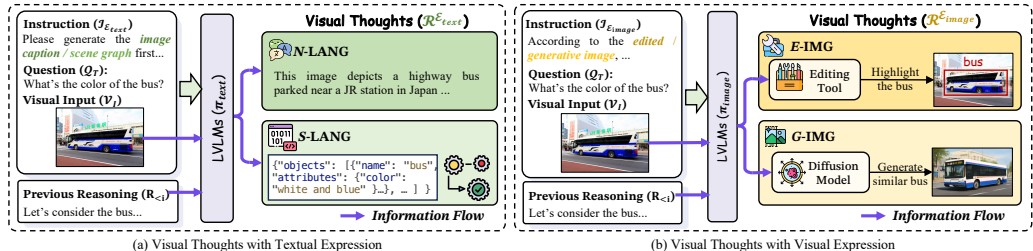

(a) Visual Thoughts with Textual Expression  (b) Visual Thoughts with Visual Expression

Figure 3: Visual Thoughts in textual expression (a) and visual expression (b). Specifically, the textual expression includes N-LANG and S-LANG, while the visual expression includes E-IMG and G-IMG.

alignment in LVLMs through richer visual descriptions [2, 12, 26], as shown in Figure 3 (a). Its reasoning process can be formally defined as:

$$\mathcal{R}_i = \underset{\mathcal{R}^{N\text{-LANG}}}{\arg\max} \pi_{text}(\mathbf{R}_{<i}, \mathcal{V}_I \rightarrowtail \mathcal{R}^{N\text{-LANG}} | \mathcal{V}_I, \mathcal{Q}, \mathcal{I}_{N\text{-LANG}}, \mathbf{R}_{<i}). \quad (3)$$

To implement $N$-LANG, we prompt LVLMs to generate image captions as a precursor to reasoning.

***Expression 2:* Structured Language ($S$-LANG)** [5] has demonstrated superior performance over traditional MCoT reasoning on math, by effectively incorporating structures language into reasoning pipelines [38, 13, 16], as shown in Figure 3 (a), which can be formally expressed as:

$$\mathcal{R}_i = \underset{\mathcal{R}^{S\text{-LANG}}}{\arg\max} \pi_{text}(\mathbf{R}_{<i}, \mathcal{V}_I \rightarrowtail \mathcal{R}^{S\text{-LANG}} | \mathcal{V}_I, \mathcal{Q}, \mathcal{I}_{S\text{-LANG}}, \mathbf{R}_{<i}). \quad (4)$$

To investigate $S$-LANG visual thoughts as a medium for expressing visual thoughts, we implement it by prompting LVLMs to generate a scene graph from input queries, which is then used for reasoning.

### 2.2.2 Interleaved Multimodal Chain-of Thought (I-MCoT)

We then introduce the MCoT through visual expressions, suggesting that image tokens are integral to visual thoughts. As depicted in Figure 1 (b), the I-MCoT framework expands on the traditional T-MCoT by integrating image editing and generation into the reasoning process, thus enabling visual thoughts to be conveyed through images. This can be mathematically represented as:

$$\mathcal{R}_i = \underset{\mathcal{R}^{\mathcal{E}_{image}}}{\arg\max} \pi_{image}(\mathbf{R}_{<i}, \mathcal{V}_I \rightarrowtail \mathcal{R}^{\mathcal{E}_{image}} | \mathcal{V}_I, \mathcal{Q}, \mathcal{I}_{\mathcal{E}_{image}}, \mathbf{R}_{<i}), \quad (5)$$

where $\pi_{image}(\cdot)$ denotes the probability of the model incorporating an image-based rationale step.

***Expression 3:* Edited Image ($E$-IMG)** processes the original image and performs various visual operations, such as grounding [23], depth estimation [48], and segmentation [35]. By conveying image tokens, $E$-IMG enhances the ability of LVLMs to interpret visual data, thereby improving reasoning capabilities, as shown in Figure 3 (b), defined as:

$$\mathcal{R}_i = \underset{\mathcal{R}^{E\text{-IMG}}}{\arg\max} \pi_{image}(\mathbf{R}_{<i}, \mathcal{V}_I \rightarrowtail \mathcal{R}^{E\text{-IMG}} | \mathcal{V}_I, \mathcal{Q}, \mathcal{I}_{E\text{-IMG}}, \mathbf{R}_{<i}), \quad (6)$$

To explore the $E$-IMG visual thought, we provide LVLMs with edited images using vision tools, enabling them to incorporate the edited results into subsequent reasoning.

***Expression 4:* Generative Image ($G$-IMG)** are required to prompt generative models to generate logical-related image based on the advancement of LVLMs [1, 9, 43, 8], as shown in Figure 3 (b), which is defined as:

$$\mathcal{R}_i = \underset{\mathcal{R}^{G\text{-IMG}}}{\arg\max} \pi_{image}(\mathbf{R}_{<i}, \mathcal{V}_I \rightarrowtail \mathcal{R}^{G\text{-IMG}} | \mathcal{V}_I, \mathcal{Q}, \mathcal{I}_{G\text{-IMG}}, \mathbf{R}_{<i}), \quad (7)$$

To explore $G$-IMG, we use DALL-E 3 [1] as a tool for visual thought to generate novel images based on input queries, which are then employed as supplementary inputs to assist reasoning.

## 3 Effectiveness Verification of Visual Thought

***Integrating visual thoughts is essential for MCoT's effectiveness in both image and text expression.***
To validate this, as depicted in Figure 4 (a), we compare system performance under three conditions:
(1) "Image-form visual thoughts", the original I-MCoT with interleaved image-and-text rationales;

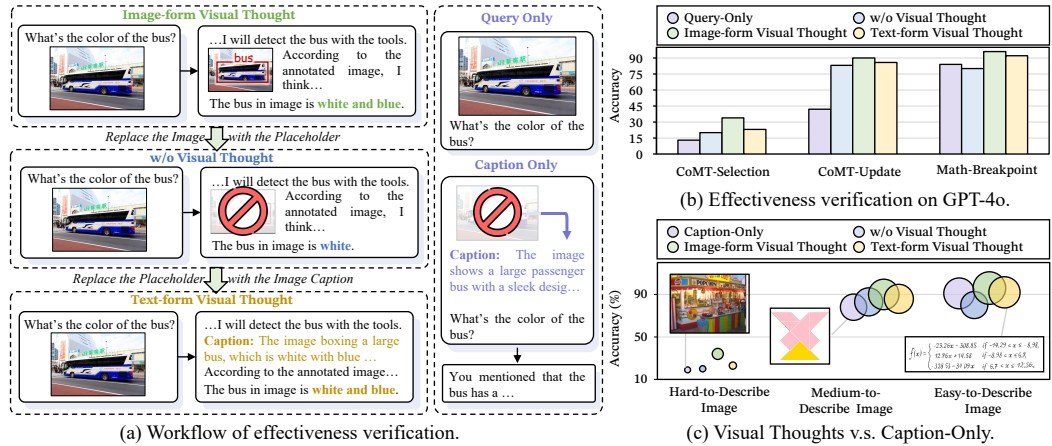

(a) Workflow of effectiveness verification.

(b) Effectiveness verification on GPT-4o.

(c) Visual Thoughts v.s. Caption-Only.

Figure 4: Effectiveness Verification for Visual Thoughts. More details are in Appendix B.

(2) "w/o visual thoughts", where the visual thought cache is cleared, forcing the model to reanalyze the input image; (3) "Text-form visual thoughts", where the cache is restored with descriptions of the images from I-MCoT. As illustrated in Figure 4 (b), the results reveal a consistent trend: omitting visual thoughts leads to a decrease in accuracy (even worse than reasoning only from the query), while including them consistently improves reasoning performance. These findings emphasize the essential role of visual thoughts in conveying visual information and enhancing model accuracy.

***Image-form visual thoughts consistently surpass text-form visual thoughts across different complexities.*** Owing to the inherent strength of the image modality in conveying visual information, image-form visual thought can facilitate cache-like visual logic propagation more effectively than text-form visual thought, thereby triggering the subsequent reasoning process with higher efficiency. As shown in Figure 4 (b), the results reveal that image-form visual thought consistently outperforms text-form visual thought across scenarios of varying complexity, with especially higher improvement pronounced in hard scenarios (47.83% on CoMT-Selection). This demonstrates that the image-form visual thought offers a superior channel for conveying detailed visual information.

***Visual thoughts represent a reasoning process that conveys visual information beyond simple image captions.*** Unlike vanilla captions, visual thoughts provide a more cache-like function that integrates detailed visual cues dynamically during reasoning. They evolve through sequential reasoning steps, encompassing visual information and contextual logic to support downstream tasks. As shown in Figure 4 (c), in simple scenes, models with visual thoughts perform comparably to caption-only baselines. However, as scene complexity increases, caption-only models lose accuracy, falling to the "w/o visual thoughts". Further, when brief captions omit essential details, visual thoughts improve performance by over 7%, showing their strength in visual information conveyance for MCoT.

## 4 Exploration for Different Categories of Visual Thoughts

Building on the effectiveness of visual thoughts observed during preliminary verification, we will further provide a comprehensive evaluation of the four classic expressions of visual thoughts[3].

### 4.1 Does these different Visual Thoughts all works?

***Explicitly incorporating visual thoughts with different expressions can all enhance the performance of almost all LVLMs.*** As shown in the results in Table 1, compared to the w/o VT expression without extra instructions, four strategies incorporating visual thoughts in the rationale achieve performance improvements across different tasks across almost all LVLMs, demonstrating the effectiveness of visual thoughts in enhancing the performance of MCoT.

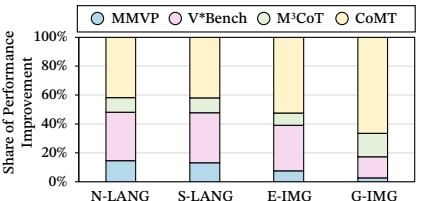

Figure 5: The proportion of performance improvement rate across tasks.

---

[3]More details can be seen in Appendix C.

| Model | MMVP | V*Bench | | M³CoT | | | CoMT | | | AVG. |
|---|---|---|---|---|---|---|---|---|---|---|
| | | position | attributes | physical | social | temporal | deletion | selection | update | |
| | | | | *LLaVA-1.5-7B* [21] | | | | | | |
| w/o VT | 45.00 | 43.42 | 29.57 | 44.44 | 59.50 | 26.83 | 21.00 | 16.00 | 23.50 | 34.36 |
| N-LANG | **52.33** | 52.63 | 34.78 | 46.67 | 60.33 | 32.52 | 21.50 | 17.50 | 29.00 | 38.58 |
| S-LANG | 51.33 | **52.63** | 35.65 | 51.11 | 61.57 | 31.71 | 22.00 | 20.50 | 29.00 | 39.50 |
| E-IMG | 49.33 | 50.00 | **36.52** | 48.89 | **64.05** | 34.15 | 25.50 | 23.00 | 29.50 | 40.10 |
| G-IMG | 49.67 | 48.68 | 34.78 | **55.56** | 63.22 | **39.02** | **29.50** | **25.00** | **35.00** | **42.27** |
| | | | | *Qwen2-VL-7B* [42] | | | | | | |
| w/o VT | 70.00 | 55.26 | 68.70 | 80.00 | 75.21 | 74.80 | 26.00 | 18.00 | 37.00 | 56.11 |
| N-LANG | **71.33** | 61.84 | **73.04** | 83.33 | 79.75 | 81.30 | 28.00 | 19.57 | 40.50 | 59.85 |
| S-LANG | 71.00 | **68.42** | 70.43 | 85.56 | 78.10 | 79.67 | 28.50 | 20.00 | 42.00 | **60.41** |
| E-IMG | 71.00 | 65.79 | 72.17 | **85.56** | **80.99** | 67.48 | 28.50 | 23.50 | **45.50** | 60.05 |
| G-IMG | 65.00 | 59.21 | 51.30 | 84.44 | 80.17 | **82.93** | **29.50** | **25.50** | 44.50 | 58.06 |
| | | | | *GPT-4o-mini* [32] | | | | | | |
| w/o VT | 72.67 | 44.74 | 36.52 | 78.89 | 70.12 | 80.49 | 10.00 | 19.50 | 24.00 | 48.55 |
| N-LANG | **75.33** | 52.17 | 61.84 | 84.44 | **79.34** | 81.30 | 27.50 | 20.00 | 27.00 | 56.55 |
| S-LANG | 74.33 | **52.63** | 54.55 | 84.44 | 74.65 | 81.30 | 26.00 | 20.00 | 33.00 | 55.66 |
| E-IMG | 73.58 | 52.63 | **70.18** | 84.44 | 76.86 | 83.74 | 29.00 | **21.50** | 33.00 | **58.33** |
| G-IMG | 72.67 | 50.00 | 53.04 | **86.67** | 78.93 | **87.80** | **30.00** | 20.00 | **40.50** | 57.73 |
| | | | | *GPT-4o* [32] | | | | | | |
| w/o VT | 74.33 | 53.95 | 54.78 | 88.89 | 76.86 | 79.67 | 26.50 | 19.50 | 37.00 | 56.83 |
| N-LANG | **85.33** | 57.89 | 63.48 | 88.89 | 78.93 | 83.74 | 33.50 | 25.50 | 37.50 | 61.64 |
| S-LANG | 84.33 | **63.16** | 64.35 | 90.00 | 78.51 | 82.93 | 29.50 | 18.00 | 42.00 | 61.42 |
| E-IMG | 83.00 | 59.21 | **65.22** | 90.00 | 78.10 | 86.18 | **34.00** | **28.50** | **50.00** | **63.80** |
| G-IMG | 78.00 | 59.21 | 59.13 | **92.22** | **78.93** | 86.18 | 33.50 | 28.50 | 46.50 | 62.46 |

Table 1: Main results on various LVLMs. The **bold content** indicates the best performance within each LVLM. `w/o VT` refers to prompting LVLMs without additional visual thoughts.

***Tasks requiring more complex visual operations can benefit more from visual thoughts.*** Further analysis, as shown in Figure 5, reveals that visual thoughts yield the most substantial performance improvements in CoMT. This benchmark primarily focuses on complex multimodal operations such as visual deletion, selection, and update during reasoning, rather than on other perception tasks. These findings suggest that visual thoughts can significantly enhance the reasoning capabilities of LVLMs in complex scenarios.

***T-MCoT demonstrates superior reasoning performance on coarse-grained perception tasks, whereas I-MCoT excels in scenarios that require fine-grained visual operations.*** Furthermore, we examine the efficiency of visual thought transmission in T-MCoT and I-MCoT across different scenarios. As shown in Table 1, T-MCoT outperforms in coarse-grained perception tasks (e.g., MMVP, V*Bench-position), while I-MCoT enables more efficient transmission in fine-grained tasks (e.g., V*Bench-attributes) and in tasks demanding visual operations (e.g., M³CoT, CoMT). Consequently, we should adapt different categories of MCoT for different features of tasks.

***Visual thoughts with visual expression cost more in rationale than textual expression.*** To compare the reasoning costs of four visual thought expressions, we calculate the average number of text tokens in responses, as well as the average number of generated image tokens for each expression. As shown in Table 2, N-LANG and S-LANG have slightly higher average text token counts, particularly S-LANG, which requires extensive structured snippets. However, for E-IMG and G-IMG, the higher average number of image tokens, due to the inclusion of more images, substantially increases the reasoning burden on LVLMs, both in terms of time and expense, indicating higher reasoning costs compared to text expressions, which highly limits the reasoning efficiency.

| | # Text Token | # Image Token |
|---|---|---|
| N-LANG | 139.02 | - |
| S-LANG | 364.37 | - |
| E-IMG | 91.65 | 1,112.51 |
| G-IMG | 89.18 | 393.00 |

Table 2: Reasoning costs of different Visual Thoughts across all LVLMs and datasets. **#X**: the average number of X.

## 4.2 Which scenarios does different Visual Thoughts work better?

***N-LANG demonstrates strong performance on coarse-grained, perception-oriented reasoning tasks.*** When a rapid overview of an entire scene is needed, N-LANG uses natural language to turn visual input into high-level semantic cues and efficiently extract macro features. As shown in Table 1, in the MMVP benchmark, where identifying a salient object matters, N-LANG first spots elements like "a butterfly", guiding subsequent analysis and achieving top results in coarse-grained perception.

***S-LANG excels in reasoning about object relationships.*** S-LANG converts input images into detailed scene graphs, enabling precise modeling of both spatial and semantic relationships. As shown in Table 1 for the V*Bench-position benchmark, S-LANG not only identifies entities such as

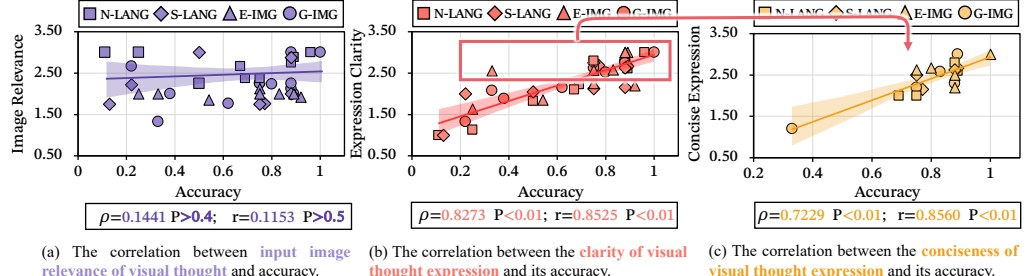

(a) The correlation between **input image relevance of visual thought** and accuracy.

(b) The correlation between the **clarity of visual thought expression** and its accuracy.

(c) The correlation between the **conciseness of visual thought expression** and its accuracy.

Figure 6: Analysis of the correlation between accuracy and visual thought quality. $\rho$: Spearman's correlation coefficient; $r$: Pearson's correlation coefficient; $P$: p-value of related assumptions.

"table" and "chair" but also accurately infers their relative positions, which underlies its state-of-the-art performance in relational reasoning tasks.

**`E-IMG` achieves strong results in detailed image analysis.** Emulating human editing workflows, `E-IMG` refines visual content to aid fine-grained feature detection. As shown in Table 1, in the V*Bench-attributes, it magnifies and annotates areas of interest, improving the accuracy of attribute predictions. This detailed focus yields the highest average performance across models.

**`G-IMG` is well-suited for multi-step reasoning through iterative image generation.** `G-IMG` generates images dynamically to refine and test reasoning hypotheses, enabling adaptive visual thinking. As shown in Table 1, in the M³CoT, requiring multiple interaction rounds, it integrates logical steps with new visuals to deepen understanding of complex concepts. This flexible generative pipeline excels in long-term, multi-turn reasoning scenarios.

### 4.3 What are the core factors that influence different Visual Thoughts effectiveness?

**Visual thoughts function as a distilled visual information cache, not a faithful replica of the original image.** A key question is whether visual thoughts accurately preserve all the content of the original image, acting as a cache instead of external storage. To explore this, we manually assess the fidelity of visual thoughts on a 0–3 scale and examine their correlation with model accuracy. As shown in Figure 6 (a), both Spearman's ($\rho$) and Pearson's ($r$) correlation coefficients are below 0.15 ($P > 0.4$), indicating no significant relationship. This suggests that visual thoughts function more as a condensed cache of visual information rather than a direct substitute for the original image.

**Visual thoughts provide a clearer, more interpretable expression of visual logic, enabling more effective retrieval of reasoning-relevant information.** Unlike direct image representations, visual thoughts do not aim to reconstruct the original image, but instead distill and communicate cross-modal visual logic like a computer cache. To assess whether the effectiveness of visual thoughts depends on how well this visual logic is expressed, we manually score the clarity of logical representation on a 0–3 scale. We then examine the correlation between these scores and model accuracy. As shown in Figure 6 (b), both Spearman and Pearson correlation coefficients exceeded 0.8 ($P < 0.01$), indicating a strong positive association. These findings suggest that the clearer the visual logical representation, the more effectively the model can reason using information stored in the visual thought cache.

**Concise visual logic expression further enhances the effectiveness of visual reasoning.** Beyond clarity, we further explore what other factors affect the effectiveness of visual thoughts. We hypothesize that more compact visual expressions, those stripped of redundant or extraneous elements—enable faster and more accurate retrieval from the internal visual cache during inference. To test this, we manually assign a conciseness score to each visual expression, reflecting the efficiency with which it conveys the visual

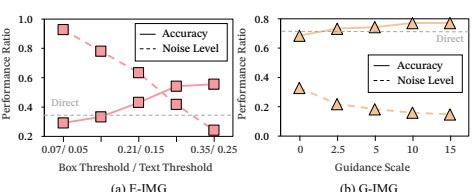

(a) E-IMG

(b) G-IMG

Figure 7: Factors Correleation of I-MCoT.

information and logic required for multimodal reasoning. We then examine how these scores relate to model performance. As shown in Figure 6 (c), reasoning accuracy correlated strongly with conciseness, which indicates that the effectiveness of visual thoughts is not solely determined by clarity, but is also enhanced by the compactness of the underlying visual logic.

**Visual thoughts affected by external noise can negatively impact the reasoning performance of MCoT.** In I-MCoT, since the visual thoughts (i.e., edited images and generative images) are typically

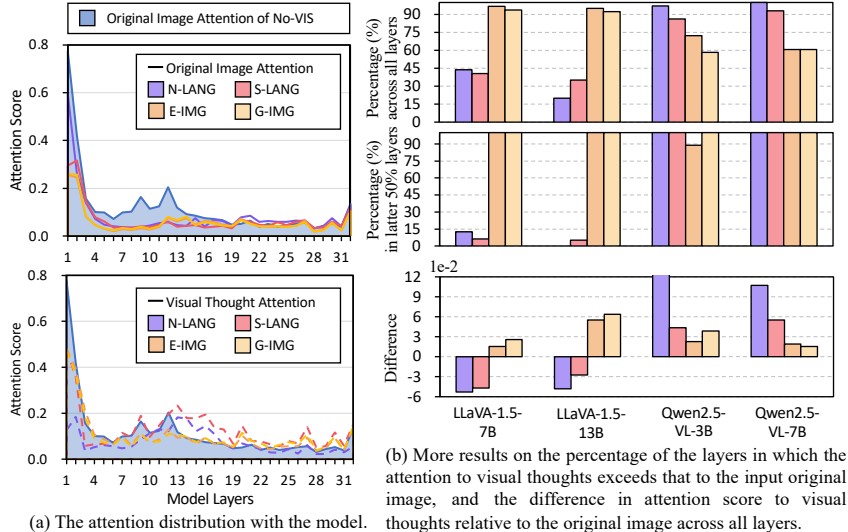

(a) The attention distribution with the model.

(b) More results on the percentage of the layers in which the attention to visual thoughts exceeds that to the input original image, and the difference in attention score to visual thoughts relative to the original image across all layers.

Figure 8: Attention distribution of Visual Thought and Image in MCoT.

produced by external tools, it is worth investigating whether the noise introduced by these tools affects MCoT's reasoning process. Specifically, we employ Grounding DINO and Stable Diffusion v1.5 to generate the edited and generative images, respectively. To control the noise introduced by external tools, we adjust their parameters to manipulate the quality of visual thoughts and evaluate the noise levels using MLLM evaluation and CLIPScore metrics. As shown in Figure 7, both E-IMG and G-IMG exhibit a clear negative correlation between accuracy and noise level. When the injected noise becomes excessive, the model's reasoning performance even fall below that of the native reasoning (Direct). This suggests that managing external noise within visual thoughts is a critical factor for maintaining effective reasoning.

## 5 Internal Rationale Behind the Visual Thought

To comprehend the rationale of visual thoughts, we examine how visual information is conveyed through attention mechanisms and information flow analysis in LVLM's internal structures.[4]

### 5.1 Visual Attention Analysis

***During the reasoning process, the attention of the original image will be scattered into visual thought.*** To investigate the internal mechanism of visual thoughts within LVLMs, we first analyze the model's attention distribution shifts of visual thoughts using LLaVA. As illustrated in Figure 8 (a), in the reasoning process without visual thought (No-VIS), the model exhibits high attention to the image, whereas in the reasoning process with visual thought, the attention to the raw input image significantly decreases across all visual thought expressions. Furthermore, we can observe that the model shifts its attention from the raw input image to various visual thoughts. This redistribution enables the flow of visual information, which supports the entire logical framework.

***Visual thoughts can convey visual information more deeply than the original image itself.*** More interestingly, as shown in Figure 8 (a), we observe a striking phenomenon in deeper layers of the model: attention to the original image without the incorporation of visual thoughts diminishes sharply after the 12th layer, nearly reaching zero. In contrast, when visual thoughts are included, attention to all expressions exhibits a marked increase, surpassing the 12th layer and even maintaining levels comparable to those seen in earlier layers. This finding suggests that visual thoughts play a crucial role in transferring visual information into the deeper layers of the model, thereby promoting enhanced cross-modal interaction and supporting more sophisticated logical reasoning.

***Model architecture has a greater impact on visual thoughts transmission than the parameter scale.*** Based on the interesting observation in Figure 8 (a), we further investigate the internal attention distribution of visual thoughts. To this end, we select the 7B(32 layers) and 13B(40 layers) model

---
[4]More details can be seen in Appendix D.

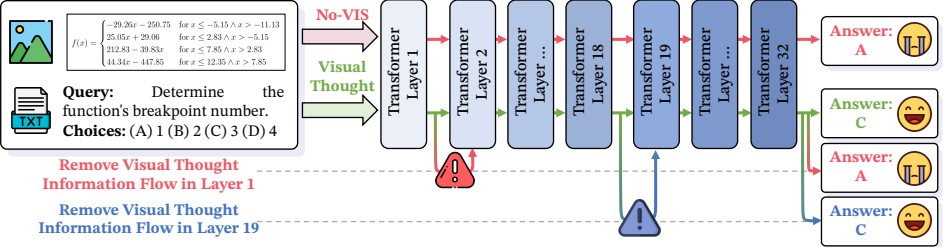

Figure 9: The Disturbation Analysis of Information Flow within LLaVA model.

from LLaVA-1.5 and select 3B(36 layers) and 7B(28 layers) model from Qwen2.5-VL series to extract internal attention to the original input image and visual thoughts seperately. As shown in Figure 8 (b), for both the LLaVA and Qwen models, in the majority of layers, the attention toward visual thoughts exceeds that toward the original image, especially in the latter 50% of layers, suggesting that *visual thoughts can convey visual information more effectively and deeply than the image itself.* Furthermore, the results of the difference in attention scores reveal that both LLaVA (7B → 13B) and Qwen (7B → 3B) exhibit increasing attention gains with deeper model layers. It reveals that *the influence of visual thoughts depends more on the model architecture (e.g., number of layers) than on the parameter scale.* Additionally, we notice that in the n-lang and s-lang settings for LLaVA, the model shows relatively low attention to visual thoughts due to the self-generated weaker language capabilities, resulting in captions and scene graphs that are overly simplistic and fail to accurately convey visual information.

## 5.2 Visual Information Flow Analysis

***Input questions query visual information from the visual thought cache.*** Subsequently, we manually disturb the model's internal information flow to understand and quantify the cache-like impact of visual thoughts on MCoT. As shown in Figure 9, without visual thoughts (No-VIS), the model erroneously selects choice A, while incorporating visual thoughts allows it to correctly choose C. Additionally, intercepting the flow of information from the query to the visual thought cache before layer 19 will significantly prevent the model from selecting the correct answer, whereas disrupting the flow from the image has no effect on the inference. These findings suggest that querying visual information from the visual thought cache is the key mechanism enhancing the model's predictions.

***The visual information retrieved from visual thought cache will be further passed to advanced reasoning processes across the boarder and deeper model layers.*** Furthermore, we examine the directional information flow distribution within the model, focusing on the flows between the the image and reasoning and the flow between visual thoughts and reasoning process at various layers. As shown in Figure 10, our findings indicate that the information flow from visual thoughts to reasoning stages is considerably stronger than the direct flow from the image to reasoning. This emphasizes the critical role visual thoughts play in mediating and organizing visual data, optimizing it for reasoning. By doing so, visual thoughts enable the model to leverage visual information more effectively, resulting in more accurate and coherent textual outputs.

***The information of the original image primarily flows to the visual thoughts.*** As illustrated in Figure 10, rather than directly entering reasoning processes, most of the image's input information is initially passed to visual thoughts during the derivation steps, which then feed into the reasoning stage. This demonstrates that nearly all image-derived signals pass through visual thoughts before reaching deeper reasoning. This two-stage process (raw pixels → visual thoughts → deeper reasoning) emphasizes the role of visual thoughts as a crucial bridge, allowing LVLMs to generate better MCoT.

## 6 Related Work

The development of chain-of-thought (CoT) [45, 17, 34] has led to research exploring its extension to multimodal reasoning [25, 44, 14, 41, 31, 53]. In this context, Zhang et al. [54] formally introduce the concept of multimodal chain-of-thought (MCoT), employing a two-stage framework that separates rationale generation and answer formulation. These studies fall under the category of Textual-MCoT (T-MCoT), where multimodal inputs are used, but the output is text-only. Zheng et al. [55], Yao et al. [49] improve multimodal interaction through step decoupling; and Wei et al. [46], Chen et al. [4] introduce multi-hop reasoning to capture more complex relationships. Furthermore, Chen et al. [6] extend T-MCoT for evaluation in commonsense reasoning tasks.

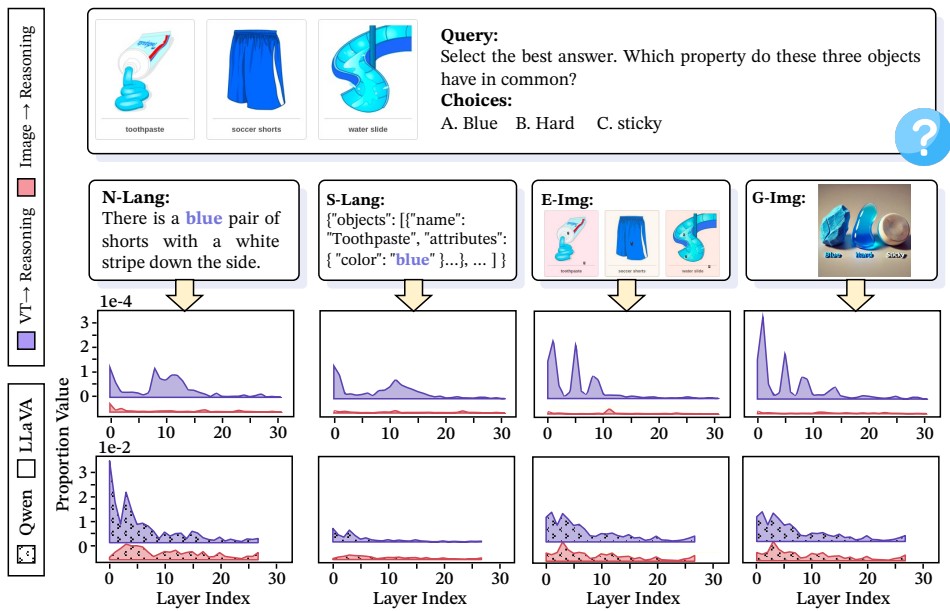

Figure 10: Information flow distribution of Visual Thought and Image in MCoT within LLaVA-1.5-7B and Qwen2-VL-2B.

Due to the limitations of T-MCoT's purely text-based output, more literature has explored MCoT using the image-text interleaved output paradigm (I-MCoT) [27, 28]. Hu et al. [15], Zhou et al. [56], Su et al. [37] explore using vision expert models to annotate input images for I-MCoT. Additionally, Cheng et al. [7] propose a benchmark of four task types requiring multimodal output to assess LVLMs' I-MCoT reasoning capabilities. While most works rely on external executors or vision expert models, Gao et al. [11] extract image patches directly from raw inputs for interleaved images, and Li et al. [19] achieve I-MCoT reasoning by fine-tuning LVLMs with multimodal generation capabilities.

Despite the advancement in T-MCoT and I-MCoT, few studies analyze the underlying unified mechanisms behind these improvements by MCoT. We attribute these performance enhancement to visual thoughts, the reasoning steps that convey visual information from the input images. We further propose four strategies that represent different modalities and approaches to integrate visual thoughts and provide a comprehensive analysis on the four strategies respectively, hoping that the findings can provide practical and systematic guidance for future research on MCoT.

## 7 Discussion

**Limitations** To facilitate variable control and streamline analysis, we exclude multiple rounds of visual thought interactions, which should be explored in future complex scenarios. Additionally, due to the difficulty in generating I-MCoT during the Any-to-Any LVLM inference process, which often leads to poor logic quality, we instead use DALLE to call for G-IMG.

**Broader Impacts** This work represents the first comprehensive investigation into the unified mechanisms underlying MCoT. We hope our work can serve as a valuable reference and guide future research into the mechanisms of MCoT. For social impact, this work may contribute to advancing the understanding of interpretability in multimodal AGI systems.

## 8 Conclusion

In this study, we first propose the concept of visual thoughts, which facilitate the transfer of visual information from input images to the reasoning process and deeper transformer layers. Visual thoughts are crucial for the effectiveness of MCoT approaches. We also propose and evaluate four strategies for expressing visual thoughts, demonstrating their role in enhancing MCoT performance. Furthermore, we analyze the underlying rationale of visual thoughts to better understand their functioning. We aim for this work to advance the understanding of the unified mechanism behind MCoT and inspire further innovations in future research.

# 9 Acknowledgements

This work was supported by the National Natural Science Foundation of China (NSFC) via grant 92570120 and 62306342. This work was supported by the Scientific Research Fund of Hunan Provincial Education Department (24B0001). This work was sponsored by the Excellent Young Scientists Fund in Hunan Province (2024JJ4070), the Science and Technology Innovation Program of Hunan Province under Grant 2024RC3024 and CCF-Zhipu Large Model Innovation Fund (NO.CCF-Zhipu202406). This study was also funded by the Open Project of the Text Computing and Cognitive Intelligence Ministry of Education Engineering Research Center (No. TCCI250101).

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

# Appendix

## A  Experiment Setup

**Model Settings**   We conduct all our experiments using four models, including *LLaVA-1.5* [21], *Qwen2-VL* [42], *GPT-4o-mini* [32], and *GPT-4o* [32]. For the GPT series models, we adjust the temperature parameter within [0,2]; for the open-source models, we adjust the temperature parameter within the range [0,2]. In addition, all open source models complete inference on 2 A6000 48G.

**Benchmark Settings**   We select benchmarks from both math and commonsense categories. For the math tasks, we choose IsoBench [10] involving tasks such as chess, math, graph, etc. For the commonsense tasks, we select datasets including MMVP [39], V*Bench [47], M3CoT-Commonsense [4], and CoMT [7], which assess the LVLMs' capabilities such as visual grounding and object detection, fine-grained identification, and CoT reasoning.

## B  Effectiveness Verification of Visual Thought

In this experimental section, we utilize the CoMT [7] and IsoBench [10] datasets.

### B.1  Visual Thought Setting

For the generation of ***Image-form Visual Thought***, for CoMT, we utilize the interleaved-modal rationales provided within the benchmark itself to construct a reasoning process that incorporates image-form visual thought, which is then provided to the model as context for answer prediction; for IsoBench, we leverage the function graphs included in the dataset as the image-form visual thought, embedding them into a template-based reasoning process that serves as context.

Subsequently, we construct two variants by modifying the generated image: in the ***w/o Visual Thought*** setting, the image is replaced with an <image> placeholder; in the ***Text-form Visual Thought*** setting, the image is substituted with its corresponding caption according to the query, generated by GPT-4o [32].

For the ***Caption Only*** setting, we employ GPT-4o [32] to generate textual descriptions directly from the original input image without consider the questions. These descriptions are then provided to the model as context for further answer prediction.

### B.2  Image Classification

In this section we investigate how the complexity of an image's textual description influences the effectiveness of visual reasoning. We first prompt the model to produce captions that are as precise as possible, and we quantify description difficulty by the number of tokens in each caption. Consistent with commonsense expectations and our manual annotations, images depicting spot-the-difference scenarios are the most challenging to describe and are therefore labeled Hard-to-Describe. Tangram-related images present a moderate level of difficulty and are labeled Medium-to-Describe. By contrast, images showing mathematical functions detected through optical character recognition (OCR) require the fewest tokens and are labeled Easy-to-Describe.

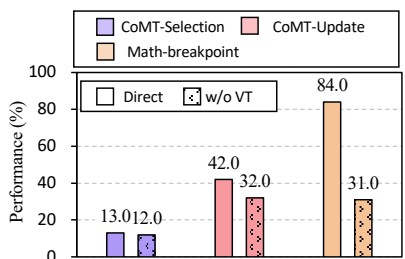

Figure 11: Results of Direct and w/o Visual Thought prompting on GPT-4o.

### B.3  Additional Discussion on w/o Visual Thoughts

To better substantiate that the performance degradation observed in the w/o Visual Thought scheme proposed in Figure 4 (a) does not merely stem from contextual perturbations, we compare direct prompting with GPT-4o to generate responses (Direct) against explicitly instructing it to avoid producing any visual content (w/o VT prompting). As shown in Figure 11, the results obtained by w/o VT prompting demonstrate a noticeable performance drop when visual information is entirely omitted by the model, even though the surrounding context remains complete.

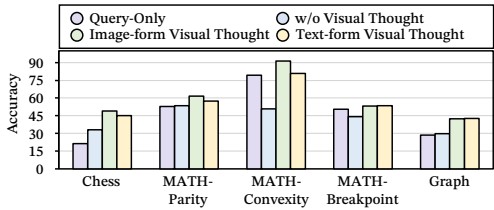
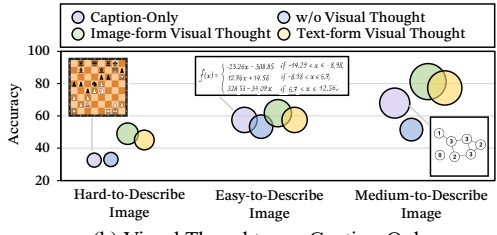

(a) Effectiveness verification on Qwen2-VL.

(b) Visual Thoughts v.s. Caption-Only.

Figure 12: Effectiveness Verification for Visual Thoughts on pure text problem.

## B.4 More Results

Beyond the exploration in Section 3, we further investigate the effectiveness of incorporating visual thought into pure-text problems that require spatial imagination. In this setting, visual thought serves to convey visual information from the imagined spatial domain that is necessary for solving the textual problems. Specifically, we select tasks including chess, function, and graph from IsoBench and use Qwen2-VL as the experimental model. We employ VisualSketchpad[15] to generate reasoning processes that include image-form visual thought, keeping all other settings the same.

***Integrating visual thoughts is essential for MCoT's effectiveness in both image and text expression.*** As shown in Figure 12 (a), consistent with the conclusions in Section 3, the absence of visual thoughts results in a noticeable drop in accuracy (even worse than relying solely on the query-only). This highlights the critical function of visual thoughts in transmitting visual information and boosting model performance.

***Visual thoughts can encode visual information more efficiently than captions in more complex scenarios.*** As shown in Figure 12 (b), in simple scenes, where descriptions are straightforward, the gain is modest (7.24%). In scenes of moderate complexity, the gain rises to 19.54%. In highly complex images, where concise captions often struggle, visual thoughts deliver over 50% efficiency improvement. These findings demonstrate that visual thoughts scale with image complexity and offer a superior channel for transmitting detailed visual information.

## C  The Categories of Visual Thoughts

In this part of the experiment, we evaluate the performance of the four types of visual thoughts using four models, including *LLaVA-1.5* [21], *Qwen2-VL* [42], *GPT-4o-mini* [32], and *GPT-4o* [32].

### C.1  Prompt Design

In the `w/o VT` setting, the model directly generates both the reasoning path and the final answer based solely on the given query. In contrast, for the four visual thought paradigms (`N-LANG`, `S-LANG`, `E-IMG`, and `G-IMG`), a two-stage framework is employed. In the first stage, the model is prompted to generate a corresponding visual thought. In the second stage, this visual thought, combined with the original query, serves as additional context to guide the model in deriving the reasoning path and the final answer.

### C.1.1  Prompt Design for `w/o VT`

In the `w/o VT` setting, the model is tasked with generating both the reasoning path and the final answer directly from the given query, without relying on intermediate visual representations. To ensure minimal influence from visual reasoning, we explicitly instruct the model to avoid referencing or incorporating any visual descriptions during the reasoning process. The specific prompts used in our experiments are presented below:

> **### Question**: *<Q>*
> **### Choices**: *<C>*
> Let's think step by step, but try to avoid any visual descriptions during the process!
> End your thinking process with the most appropriate answer in the format "ANSWER: (x)"
> followed by the choice.
> Your Response:

### C.1.2 Prompt Design for `N-LANG`

`N-LANG` facilitates effective visual information transfer by natural language expression, such as describing images based on question, enhancing vision-language alignment in LVLMs through richer visual descriptions.

**Stage 1: Visual Thought Generation**    At this stage, the model is tasked with extracting meaningful information and transforming it into natural language, forming a foundation for visual thought and enabling downstream reasoning. The specific prompts used in our experiments are as follows:

> **### Question**: *<Q>*
> **### Choices**: *<C>*
> Please generate a comprehensive caption for the given image based on the provided query,
> ensuring it accurately reflects the content and context of the image and the query.
> **### Caption**:

**Stage 2: Visual Thought Reasoning**    Then, the model advances to cross-modal reasoning, building on its previously generated visual thoughts. Below are the prompts used in this stage of our experiment:

> Based on the question and the caption that is related to the question and generated by yourself,
> let's think step by step, but try to avoid adding visual descriptions during the process! End your
> thinking process with the most appropriate answer in the format "ANSWER: (x)" followed by
> the choice.
> **### Question**: *<Q>*
> **### Choices**: *<C>*
> **### Caption**: *<N-LANG>*
> Your Response:

### C.1.3 Prompt Design for `S-LANG`

`S-LANG` facilitates effective visual information transfer by effectively incorporating structures language into reasoning pipelines.

**Stage 1: Visual Thought Generation**    In this stage, the provided image constitutes the primary source of visual information. The model is required to carefully analyze the visual content and interpret it in a structured manner that facilitates subsequent reasoning processes. The specific prompts employed in our experiments are as follows:

> **### Question**: *<Q>*
> **### Choices**: *<C>*
> For the provided image and its associated question, generate a scene graph in JSON format
> that includes the following:
> 1. Objects that are relevant to answering the question.
> 2. Object attributes that are relevant to answering the question.
> 3. Object relationships that are relevant to answering the question.

**Stage 2: Visual Thought Reasoning**   Next, the model performs advanced cross-modal reasoning based on the visual thoughts generated in the previous stage. The specific prompts used in our experimental setup are listed below:

> Based on the question and the scene graph that is related to the question and generated by yourself, let's think step by step, but try to avoid adding visual descriptions during the process! End your thinking process with the most appropriate answer in the format "ANSWER: (x)" followed by the choice.
> **### Question**: *<Q>*
> **### Choices**: *<C>*
> **### Scene Graph**: *<S-LANG>*
> Your Response:

### C.1.4 Prompt Design for `E-IMG`

`E-IMG` transforms the original image through tasks such as grounding, depth estimation, and segmentation. By encoding image tokens, it strengthens LVLMs' capacity to interpret visual information, thereby enhancing reasoning performance.

**Stage 1: Visual Thought Generation**   At this stage, `E-IMG` employs vision tools (e.g., Grounding-DINO [24], Semantic-SAM [20], DepthAnything [48]) to edit images based on the *action series*. These processed images serve as extended inputs, facilitating the formation of visual thoughts in the subsequent reasoning stage. The prompts used in our experiments are as follows:

> **### Question**: *<Q>*
> **### Choices**: *<C>*
> Given the image, questions, and options, please think step-by-step about the image to answer the question, design a series of image processing steps to extract pointing visual features based on the available actions.
> The available actions are:
> 1. segment_and_mark():
> 2. detection(objects):
> ...
> You are encouraged to use as few steps as possible to achieve the goal.
> **# # # Action Series**:

**Stage 2: Visual Thought Reasoning**   At this stage, the model advances to cross-modal reasoning tasks, guided by visual thought representations formed earlier. The prompts used in our experimental procedure are detailed below:

> Based on the question and the additional annotated image (all images except the first one) that is related to the question and created according to Image Processing Series generated by yourself, let's think step by step, but try to avoid adding visual descriptions during the process! End your thinking process with the most appropriate answer in the format "ANSWER: (x)" followed by the choice.
> **### Question**: *<Q>*
> **### Choices**: *<C>*
> *<Extra Image Input>*
> Your Response:

### C.1.5 Prompt Design for `G-IMG`

`G-IMG` is designed to prompt generative models to produce images that support logical reasoning, leveraging the progress of LVLMs.

| Model | MMVP | V*Bench | | M³CoT | | | CoMT | | | AVG. |
|---|---|---|---|---|---|---|---|---|---|---|
| | | position | attributes | physical | social | temporal | deletion | selection | update | |
| w/o VT | 74.33 | 53.95 | 54.78 | 88.89 | 76.86 | 79.67 | 26.50 | 19.50 | 37.00 | 56.83 |
| N-LANG | 85.33 | 57.89 | 63.48 | 88.89 | 78.93 | 83.74 | 33.50 | 25.50 | 37.50 | 61.64 |
| S-LANG | 84.33 | 63.16 | 64.35 | 90.00 | 78.51 | 82.93 | 29.50 | 18.00 | 42.00 | 61.42 |
| E-IMG | 83.00 | 59.21 | 65.22 | 90.00 | 78.10 | 86.18 | 34.00 | 28.50 | 50.00 | 63.80 |
| G-IMG | 78.00 | 59.21 | 59.13 | 92.22 | 78.93 | 86.18 | 33.50 | 28.50 | 46.50 | 62.46 |
| w/o CoT | 77.67 | 57.89 | 63.48 | 87.78 | 76.03 | 58.54 | 32.50 | 21.50 | 41.50 | 57.43 |

Table 3: Results of w/o CoT on GPT-4o.

**Stage 1: Visual Thought Generation**   At this stage, `G-IMG` employs the vision model DALL·E 3 [1] to generate images based on tailored *image generation prompts*. These images embody the model's visual thoughts and are used as additional inputs in the subsequent reasoning phase. The prompts used in our experiments are as follows:

> You are an expert in writing prompts for text-to-image generation.
> Now, based on the following image and the corresponding textual query, please write a precise and detailed prompt to generate an image that is highly relevant to the query.
> This image will be provided to you later as an auxiliary tool to help answer the query. Therefore, the generated image should be as clear, detailed, and closely aligned with the query as possible, helping you extract the necessary information from the image to answer or resolve the query accurately.
> When writing the prompt, please consider factors such as the composition, color, style, and details of the image to ensure its practicality and effectiveness.
> **### Question**: *<Q>*
> **### Choices**: *<C>*
> **### Prompt Generated**:

**Stage 2: Visual Thought Reasoning**   In this phase, the model performs cross-modal reasoning tasks, drawing on visual thought representations established in the prior stage. The corresponding experimental prompts are presented below.

> Based on the question and the additional synthesized image (the second one) that is related to the question, let's think step by step, but try to avoid adding visual descriptions during the process! End your thinking process with the most appropriate answer in the format "ANSWER: (x)" followed by the choice.
> **### Question**: *<Q>*
> **### Choices**: *<C>*
> ***<Extra Image Input>***
> Your Response:

## C.2   More Results on `w/o CoT`

To gain a deeper understanding of the role of visual thoughts, we supplement the results in Table 1 with an additional vanilla baseline which is without any CoT reasoning, implemented using GPT-4o. As shown in the Table 3, we observe that the trends of removing Visual Thoughts (VT) and CoT are not consistent. For benchmarks that require more reasoning steps, such as M³CoT, which involves extensive multi-step reasoning, the performance of the w/o CoT variant significantly degrades.

## C.3   Core Factors Evaluation Mechanism

In Section 4.3, we conduct a human evaluation to assess the conciseness and efficiency of visual thoughts. Specifically, we randomly sample 100 instances evenly distributed across MMVP [39], V*Bench [47], M3CoT [4], and CoMT [7].

To evaluate the effectiveness of visual thoughts generated in response to user queries, we introduce three evaluation metrics. Each metric is designed to capture a specific aspect of visual reasoning quality, as defined below:

- **Image Relevance**: This metric assesses the degree to which the visual thoughts aligns with the semantic content and context of the input image. A highly relevant visual thoughts output should accurately reflect the objects, relationships, or scenes depicted in the image, ensuring that the generated content remains faithful to the source material.

- **Expression Clarity**: This criterion measures how clearly the visual thoughts conveys the intended reasoning or logic in response to the input query. A high score indicates that the visual elements, such as spatial arrangements, symbols, or visual cues, are intuitively understandable and unambiguous, allowing viewers to easily grasp the underlying rationale.

- **Concise Expression**: This metric evaluates the efficiency and succinctness of the visual thoughts in communicating information. An effective visual thoughts should avoid unnecessary visual complexity while preserving essential content, thereby enhancing interpretability and reducing cognitive load on the viewer.

Each of the three metrics is scored on a 3-point ordinal scale: (1) *Low*: The visual thoughts fails to meet the criterion or introduces confusion. (2) *Medium*: The visual thoughts partially meets the criterion, with minor ambiguities or limitations. (3) *High*: The visual thoughts fully satisfies the criterion in a clear, accurate, and efficient manner.

Detailed annotation guidelines are provided as below:

---

Each metric is rated on a 3-point scale: *low* (1), *medium* (2), and *high* (3). The evaluation criteria for each score level are detailed below:

**Image Relevance:** It measures the consistency between the visual thought and the content of the input image.
- *Low*: The visual thought misrepresents or incorrectly describes the image content.
- *Medium*: The visual thought accurately captures the main content of the image.
- *High*: The visual thought not only accurately represents the image but also provides a comprehensive description.

**Expression Clarity:** Assesses how clearly the visual thought represents the visual logic implied by the input query.
- *Low*: The visual thought fails to convey any visual logic relevant to the query.
- *Medium*: The visual thought partially captures the visual logic relevant to the query.
- *High*: The visual thought fully and clearly expresses the visual logic associated with the query.

**Concise Expression:** Evaluates the comprehensibility and succinctness of the visual thought in conveying visual information.
- *Low*: The visual thought is verbose, redundant, or difficult to understand.
- *Medium*: The visual thought conveys its visual content and logic in a generally clear manner.
- *High*: The visual thought presents the visual content and logic clearly, concisely, and in an easily understandable way.

---

## D   Internal Rationale Behind the Visual Thought

In Section 5, we analyze the internal mechanisms by which MCoT transmits visual information, focusing on two perspectives: the attention mechanism and information flow.

| Model | MMVP | V*Bench | | M³CoT | | | CoMT | | | AVG. |
|---|---|---|---|---|---|---|---|---|---|---|
| | | position | attributes | physical | social | temporal | deletion | selection | update | |
| w/o VT | 74.33 | 53.95 | 54.78 | 88.89 | 76.86 | 79.67 | 26.50 | 19.50 | 37.00 | 56.83 |
| N-LANG(Maj@4) | 85.00 | 57.89 | 63.48 | 88.89 | 78.93 | 83.74 | 33.50 | 25.50 | 37.50 | 61.60 |
| S-LANG(Maj@4) | 83.67 | 63.16 | 64.35 | 90.00 | 78.51 | 82.93 | 29.50 | 18.00 | 42.00 | 61.35 |
| E-IMG(Maj@4) | 82.33 | 59.21 | 65.22 | 92.22 | 78.10 | 86.18 | 33.50 | 28.00 | 49.00 | 63.75 |
| G-IMG(Maj@4) | 78.00 | 59.21 | 63.48 | 92.22 | 78.93 | 86.18 | 33.50 | 28.50 | 46.50 | 62.95 |
| Diverse-VT | **85.00** | **63.16** | **65.22** | **92.22** | **78.93** | **86.18** | **34.00** | **28.50** | **50.00** | **64.80** |

Table 4: Results of Diverse-VT on GPT-4o.

## D.1 Visual Attention Analysis

In this section, we employ *LLaVA-1.5-7B* [21] as the experiment model to analyze the attention distribution during its reasoning process and randomly select one sample from the ScienceQA [25] as a case study. Specifically, we compare two input settings: (1) *a multimodal query alone*, and (2) *a multimodal query with visual thought*. For each setting, we extract the attention weights from the last token to the input image and the visual thought in each transformer layer.

We define $\mathbf{A}_{i,j}^{(l,h)}$ as the attention weight of token $i$ attending to token $j$ of the $h$-th head at the $l$-th layer, and the attention weight extraction process can be represented as:

$$\bar{a}_{\text{last} \to k}^{(l)} = \frac{1}{H} \sum_{h=1}^{H} \mathbf{A}_{\text{last},k}^{(l,h)},$$ (8)

where $\bar{a}_{\text{last} \to k}^{(l)}$ represented the target extracted attention weight, $last$ represents the index of the last token, $k$ represents the index of the target token, and $H$ represents the number of the head.

As illustrated in Figure 8, we observe that the inclusion of visual thoughts leads to a shift in attention from the original image toward the visual thought. Moreover, the visual thought continues to carry and transmit visual information into deeper layers of the transformer.

## D.2 Visual Information Flow Analysis

In this section, we also utilize *LLaVA-1.5-7B* [21] as the experimental model and sample data from the ScienceQA [25] to investigate the role of internal information flow within MCoT. Specifically, we conduct two analyses: Attention Blocking Analysis and Saliency-Based Information Flow Analysis

### D.2.1 Attention Blocking Analysis

We manually block the information flow between specific tokens by setting the attention mask between them in selected transformer layers to `-inf` [40, 50]. As shown in Figure 9, blocking the information flow between the query and the visual thought leads the model to choose an incorrect option that it would otherwise have predicted correctly. This result highlights the critical influence of information flow between the query and the visual thought on final answer prediction.

### D.2.2 Saliency-Based Information Flow Analysis

We compute saliency scores [36] to evaluate the relative importance of different information flows for answer prediction [40, 50]. The computing process utilize Taylor expansion [29] for each element of the attention matrix:

$$I_l = \left| \sum_h A_{h,l} \odot \frac{\partial \mathcal{L}(x)}{\partial A_{h,l}} \right|,$$ (9)

Here, $\mathcal{L}(x)$ represents the loss function, and $x$ denotes the input; $A_{h,l}$ is the attention matrix value for the $h$-th attention head in the $l$-th layer. The saliency matrix $I_l$ for the $l$-th layer is obtained by averaging across all heads. The significance of information flow from the $j$-th token to the $i$-th token can be represented by $I_l(i,j)$.

As illustrated in Figure 10, the information flow between the visual thought and the reasoning process plays a pivotal role in guiding the model toward the correct answer, and its contribution is significantly greater than that of the image-to-reasoning pathway.

# E    Future Work

Based on the empirical analysis of visual thoughts, exploring ways to further enhance the performance of MCoT is a promising direction for future research. Here, inspired by self-consistency, we design a novel visual-thought-guided strategy that ensembles diverse visual thoughts reasoning paths, which are refered as diverse-VT. We implement diverse-VT on GPT-4o. As shown in Table 4, we observe that Diverse-VT achieves the best performance, demonstrating the feasibility of integrating multiple visual thoughts to enhance visual information transmission in MCoT, thereby improving reasoning performance.

# F    Ethical Considerations

We engage multiple human annotators to evaluate the quality of visual thoughts in Section 4.3.

**Quality Check**    We initiated the project with an introductory interview task, in which participants answered 10 example questions for each visual thought expression. To ensure participant engagement and familiarize them with the task, each participant was compensated $20.

**Dataset Annotation**    For the subsequent stages of data annotation, we employed two PhD students and two graduate students who possessed proficiency in both Chinese and English (CET-6 level) and demonstrated strong mathematical capabilities. These students were compensated $15 per hour, which is higher than the local average salary.

