# OpenReview forum: "Visual Thoughts: A Unified Perspective of Understanding Multimodal Chain-of-Thought"
_NeurIPS.cc/2025/Conference — NeurIPS 2025 poster_

### Official Review · Reviewer_79f4 · 2025-06-22

**Clarity:** 3
**Significance:** 2
**Originality:** 3
**Rating:** 4
**Confidence:** 3

**Summary:**

Despite advancements in T-MCoT and I-MCoT, few studies analyze the underlying unified mechanisms behind these improvements in MCoT. The paper attributes these performance enhancements to visual thoughts, the reasoning steps that convey visual information from the input images.  The authors further propose four strategies that represent different modalities and approaches to integrate visual thoughts, providing a comprehensive analysis of each strategy.

**Questions:**

1. The experiments primarily use LLaVA-1.5-7B for attention and information flow analyses, and it is unclear how generalizable the findings are across different LVLM architectures or scales. Could the authors extend their analysis to at least one additional LVLM with a different architecture or scale to validate the robustness of visual thoughts?
2. The authors claim that removing visual thoughts and forcing reasoning solely from the original image can impair performance more than reasoning directly from the query, as demonstrated by the Math-Breakpoint results in Figure 4 (b). This finding appears counterintuitive and lacks adequate discussion. Furthermore, is it appropriate to draw such a broad conclusion based on a single experiment?
3. Image-based visual thoughts (E-IMG, G-IMG) rely on external tools such as Grounding-DINO and DALL-E 3, which may introduce inaccurate segmentations or irrelevant generated images. How do such errors impact reasoning performance, and how are they mitigated? Could the authors evaluate the sensitivity of MCoT performance to errors in visual thought generation (e.g., by injecting controlled noise or using lower-quality tools)?

**Ethical Concerns:**

["NO or VERY MINOR ethics concerns only"]

**Final Justification:**

The paper offers a thorough analysis of Visual Thoughts. Its findings have the potential to inform future research; however, the insights are somewhat limited. Therefore, I recommend a rating of Borderline Accept.

**Limitations:**

Yes.

**Paper Formatting Concerns:**

No.

**Quality:**

3

**Strengths And Weaknesses:**

**Strengths:**
1. The paper conducts a systematic evaluation of four visual thought expressions (N-LANG, S-LANG, E-IMG, G-IMG) across multiple tasks (MMVP, V*Bench, M3CoT, CoMT). This multi-task, multi-model approach strengthens the robustness of the findings by demonstrating the effectiveness of visual thoughts across diverse scenarios.
2. The internal analysis of visual thoughts (Section 5) using attention distribution (Figure 7) and information flow (Figures 8–9) within LVLMs provides a novel perspective on how visual information is processed. This adds depth to the study, as it connects empirical results to model mechanics.

**Weakness:**
1. The experiments primarily focus on a small set of LVLMs (e.g., GPT-4o, LLaVA), which limits the generalizability of the findings. Without testing on a broader range of architectures, the claim of a “unified perspective” is weakened.
2. The correlation analysis of clarity and conciseness (Figure 6) relies on manual scoring (0–3 scale). This introduces subjectivity and reduces the reproducibility of the results.
3. While Table 1 reports performance across tasks, the paper lacks a detailed discussion of why certain visual thought expressions perform better in specific tasks. For example, the superior performance of E-IMG in V* Bench-attributes is noted but not thoroughly analyzed, which limits the interpretability of the results.

---

> ### Author Rebuttal · Authors · 2025-07-31
>
> Thanks for your acknowledgment and interest in our work! We sincerely appreciate your thorough and insightful comments on our work, and we will address each of your main concerns below:
>
> **Q1**: The experiments primarily focus on a small set of LVLMs.
>
> **R1**: Thanks for your insightful comment. In our paper, we select a range of LVLMs with diverse architectures and parameter scales, spanning from proprietary models (GPT-4o, GPT-4o-mini) to open-source models (Qwen2-VL, LLaVA-1.5), in order to provide the broadest possible interpretation of the role of visual thoughts.
>
> **Q2**: The correlation analysis of clarity and conciseness (Figure 6) relies on manual scoring.
>
> **R2**: Thanks for your constructive comment. We also employ the automatic evaluation method CLIPScore to assess all the visual thoughts presented in Table 1. Specifically, we compute the classification probability between two categories: “This is a clear (annotated logical) image” and “This is not a clear (annotated logical) image” (for G-IMG, content in parentheses is removed). These probabilities serve as quantitative indicators for clarity score of visual thought.
> > Table 1: Correlation analysis between clip-based clarity scores and model performance on GPT-4o-mini.
> || Pearson Correlation| P |
> |-|:-:|:-|
> | S-LANG | 0.692| 0.06 |
> | N-LANG | 0.693| 0.04 |
> | G-IMG  | 0.731 | 0.04|
> | E-IMG  | 0.646| 0.06|
>
> From the results, we can observe that the trends from human evaluation and automatic evaluation are consistent.
>
> We will add more discussion in the next version.
>
> **Q3**: Certain sections lack a detailed discussion of why certain visual thought expressions perform better in specific tasks.
>
> **R3**: Thanks for your kind feedback. To investigate the performance differences of visual thoughts across various tasks, we analyze them in conjunction with the characteristics of specific task sets. For instance, CoMT—which places greater emphasis on visual information interaction—tends to benefit more from image-form visual thoughts that effectively convey information. In Q6, we further examine the impact of noise within visual thoughts on task-specific performance, thereby further revealing the factors that contribute to the performance gains brought by visual thoughts. We will add more discussion in the next version.
>
> **Q4**: The experiments for attention and information flow analyses is unclear how generalizable the findings are across different LVLM architectures or scales.
>
> **R4**: Thanks for your insightful feedback. We follow your suggestion to add more models for evaluation.
> 1. **For attention analysis**:
>
> We conduct attention analysis across four models (LLaVA-7B/13B and Qwen2.5-VL-3B/7B) by comparing attention to the input image (w/o visual thoughts) and to visual thoughts (with VT). According to the result, we observe that across most layers—and especially in the latter half—attention to visual thoughts consistently exceeds that to the original image, indicating that visual thoughts can more effectively convey visual information during reasoning, **consistent with findings in the paper**.
>
> Moreover, we find that models with more layers exhibit stronger attention to visual thoughts, suggesting that architecture depth is a key factor in leveraging visual thoughts.
>
> For methodological details and specific experimental results, please refer to our detailed response to **Reviewer #NrVc (R1)**.
>
> 2. **For information flow analysis**:
>
> We incorporate *Qwen2-VL-2B* to examine information flow under different types of visual thoughts. According to the experimental results, all variants of visual thoughts yield stronger information flow to reasoning stages than the original image, **consistent with our findings in LLaVA models**.
>
> Notably, unlike LLaVA, Qwen2-VL exhibits the highest information flow in the n-lang setting, likely due to its stronger language generation capabilities.
>
> For a detailed explanation and discussion, please refer to our response to **Reviewer #NrVc (R2)**.
>
> **Due to space constraints, we summarize only the key insights here. We sincerely appreciate your understanding!**
>
> We will add more discussion in the next version.
>
> **Q5**: The finding of 'forcing reasoning solely from the original image can impair performance more than reasoning directly from the query' appears counterintuitive and lacks adequate discussion.
>
> **R5**: Thanks for your valuable comment. We sincerely believe there may have been a misunderstanding regarding our explanation. Actually, what we intended to convey is that in Figure 4(b), removing the visual thought module can, in some cases, lead to performance even lower than the Query-Only baseline. However, this does not imply that all instances of the “w/o Visual Thought” condition perform worse than the Query-Only baseline (as clarified in the first paragraph of Section 3).
>
> Moreover, this phenomenon is observed only in the Math subset, primarily because many Math questions contain sufficiently detailed textual descriptions that allow the model to perform reasoning without relying heavily on visual features. Removing visual thoughts in such cases disrupts the effective transmission of visual cues, which unexpectedly degrades performance. In contrast, for other datasets, the queries themselves carry minimal information (e.g., questions such as “How many people are in the image?”), and thus the Query-Only condition naturally performs worse than “w/o Visual Thought”.
>
> To further address your concerns, we have included additional experiments, as illustrated in Table 2. Similar trends are observed not only in GPT-4o but also in other models such as GPT-4o-mini and Gemini.
> > Table 2: Results on w/o visual thought v.s. Query-Only.
> ||chess|graph|math|
> |-|:-:|:-:|:-:|
> |GPT-4o-mini w/o visual thought|6.00%|38.00% |9.00%|
> |GPT-4o-mini Query-Only|0.00%|3.00%|26.00%|
> |Gemini-2.0-flash w/o visual thought |4.00%| 24.00%|17.00%|
> |Gemini-2.0-flash Query-Only|1.00%|0.00%|29.00%|
>
> We will incorporate more detailed discussion on this issue in future versions of the paper to ensure clarity and alleviate any potential confusion.
>
> **Q6**: How do errors injected from inaccurate segmentations or irrelevant generated images impact reasoning performance, and how are they mitigated?
>
> **R6**: Thanks for your constructive feedback. To evaluate the potential performance variations in MCoT caused by errors introduced through image-form visual thoughts, we conduct the following experiment:
>
> **For E-Img**: To evaluate the potential errors introduced by E-Img, we controll the editing effects by adjusting the parameters of the editing tool and observe corresponding changes in MCoT performance. Specifically,
> 1. We select GroundingDino as the editing tool and manipulate the *box threshold* and *text threshold* parameters to vary the accuracy of grounding. Images with poor grounding mistakenly annotate irrelevant objects as grounded, resulting in inaccurate edited images.
> 2. These edited images are then ranked by GPT-4o based on grounding quality, and the resulting ranking is used to quantify the introduced errors by assigning specific scores to specific ranks.
> 3. We sample 100 instances from V*Bench-attributes which are sensitive to E-Img and remove the instance failing to perform grounding, using GPT-4o-mini as the model. The results are shown in Table 3.
> 4. We include the method without incorporating visual thoughts (no-vis) as the baseline.
>
> **For G-Img**: To facilitate the control of the quality of the G-IMG, we controll the generation effects by adjusting the parameters of the generation models and observe corresponding changes in MCoT performance. Specifically,
> 1. We adopt the open-sourced *Stable Diffusion v1.5*. We manipulate the model’s *guidance scale* parameter to control the degree to which the generated image adheres to the input prompt. Images generated with a low guidance scale often fail to reflect visual information relevant to the original query, thereby resulting in irrelevant generated images.
> 2. We then used CLIPScore to evaluate the similarity between the generated images and the input prompts, serving as a metric to quantify the introduced error.
> 3. We sample 100 instances from M3CoT which is sensitive to G-Img as the experiment data, using GPT-4o-mini as the model. The results are presented in Table 4.
> 4. We include the method without incorporating visual thoughts (no-vis) as the baseline.
> > Table 3: Evaluation of E-Img error sensitivity.
> |box threshold & text threshold(↑)|noise score(↓)|acc(↑)|
> |:-:|:-:|:-:|
> |0.07 & 0.05| 0.9278|29.17|
> |0.14 & 0.10| 0.7806|33.33|
> |0.21 & 0.15| 0.6333| 43.06|
> |0.28 & 0.20|0.4167|54.17|
> |0.35 & 0.25|0.2417| 55.56 |
> |no-vis|-|34.72|
> > Table 4: Evaluation of G-Img error sensitivity
> | guidance scale(↑) | clipscore(↑) | noise score(↓) | acc(↑) |
> |:-:|:-:|:-:|:-:|
> |0|0.6720|0.3280|68.57|
> |2.5|0.7815|0.2185|73.33 |
> |5|0.8168|0.1832|74.29|
> |10|0.8401|0.1599|77.14|
> |15|0.8519|0.1481|77.14|
> |no-vis|-|-|72.38|
>
> Based on the results in Tables 3 and 4, we observe a positive correlation between the quality of edited and generative image and the MCoT performance.
>
> We further observe that poor-quality edited images (box threshold & text threshold=0.07 & 0.05) or generated images (guidance scale=0) can even impair the reasoning capabilities of MCoT (lower than *no-vis*). Through manual inspection, we find that those images contain substantial misleading information (e.g., grounding numerous objects that do not meet the specified conditions or generating images that convey incorrect visual content). This indicates that inaccurate visual thoughts may introduce noise and, in turn, lead to degraded performance.
>
> To mitigate the introduced errors, we can adopt more powerful editing and generative models, and leverage general-purpose LVLMs to evaluate the resulting images for timely correction.
>
> We will **DEFINITELY** add more the discussion for mitigate your concerns in the next version.

---

> > ### Author Response · Authors · 2025-08-02
> >
> > Thank you very much for your time and valuable feedback. Hope our clarifications can address your concerns, and we sincerely hope that you can reconsider our work in light of these clarifications. If you have any further comments, please do not hesitate to contact us. We greatly appreciate your selíess contributions to the community.

---

> ### Comment · Reviewer_79f4 · 2025-08-04
>
> Most of the questions have been addressed, and I am considering updating my final rating. I would like to see these discussions included in the final version.

---

> > ### Author Response · Authors · 2025-08-04
> > **Thank you for your acknowledgement**
> >
> > We greatly appreciate your thoughtful feedback and helpful suggestions. We're glad to hear that our response addressed your concerns. We will **definitely** follow your suggestions to further improve our work in the next version. Thank you again for your selfless contributions to the community.

---

> ### Comment · Reviewer_79f4 · 2025-08-07
>
> I have no further questions. While I find the results valuable, the insights provided are somewhat limited. Therefore, I will maintain my current rating.

---

### Official Review · Reviewer_Z1x4 · 2025-06-30

**Clarity:** 2
**Significance:** 2
**Originality:** 3
**Rating:** 4
**Confidence:** 2

**Summary:**

The paper introduces and analyzes the concept of 'visual thoughts', intermediate reasoning steps in Multimodal Chain-of-Thought (MCoT) reasoning with Vision-Language Models (VLMs). These visual thoughts, expressed as either text or images, serve as distilled representations of visual input that differentially enhance VLM performance across tasks. Acting as a form of internal visual cache, visual thoughts reduce the need for repeated image processing and help propagate visual information into deeper layers of the model during reasoning.

**Questions:**

- What metric is plotted on the y-axis of Figure 5? Why do all columns add up to 1? Please clarify in the caption or main text.
- In Section 5.2, how exactly is the intervention performed to disrupt the information flow from visual thoughts or the image? Is this based on manipulating attention scores?

**Ethical Concerns:**

["NO or VERY MINOR ethics concerns only"]

**Final Justification:**

I acknowledge the efforts made by the authors in the rebuttal, and particularly their explanations related to the Effectiveness Verification of Visual Thoughts and the Visual Attention Analysis. While I share the view of other reviewers that the paper introduces limited technical innovation, I believe that the unified analysis of multimodal CoT introduced in the paper is valuable, and could inform future research in the field. Therefore, I am voting for borderline acceptance.

**Limitations:**

Yes

**Paper Formatting Concerns:**

No concern.

**Quality:**

2

**Strengths And Weaknesses:**

**Strengths:**

- The paper offers practical insights into the suitability of different types of CoT strategies for various tasks. These observations are well-motivated and could guide future work in selecting or designing appropriate MCoT strategies depending on task characteristics.

- The mechanistic analysis of internal model behavior, particularly attention shifts and information flow, is interesting and contributes meaningfully to our understanding of how visual reasoning unfolds in large vision-language models.

**Weaknesses:**

- **Effectiveness verification of visual thought.** I have some concerns regarding the methodology for the 'w/o visual thought' and 'text-form visual thought' baselines. If these are generated by simply replacing the image in the image-form visual thought outputs with a placeholder or caption, the surrounding text may still implicitly reference the image, which could introduce inconsistencies in the context. This may negatively impact performance not solely due to the absence of the visual thought, but because the resulting context is no longer coherent. Additionally, the authors state that removing visual thoughts leads to performance worse than using the query alone, but Figure 4b shows that the query-only setting is consistently the lowest-performing or close to it, calling this claim into question.

- **Baselines.** The 'w/o VT' baseline of Section 4 is unclear: does it correspond to the 'w/o visual thought' from Figure 4? If so, the paper should clarify this explicitly. Additionally, it would be useful to include a vanilla baseline without any CoT reasoning at all, to better isolate the value added by CoT versus visual thoughts.

- **Visual Attention Analysis.** The conclusions drawn from the attention plots are not entirely convincing. While the paper claims that visual thought tokens receive higher attention in deeper layers, the observed differences appear subtle, and the attention to visual thoughts does not clearly dominate over original attention to the image. Stronger quantitative evidence would strengthen this analysis.

**Typos:**

The manuscript contains some minor typos, including:

- line 154 'Does these different Visual Thoughts all works?' -> 'Do these different Visual Thoughts all work?'
- line 188 'Which scenarios does different Visual Thoughts work better?' -> 'In which scenarios do different Visual Thoughts work better?'
- line 176 it is unclear what 'in rationale' means in this context
- line 264 it is unclear what 'boarder' means in this context

---

> ### Author Rebuttal · Authors · 2025-07-31
>
> Thanks for your acknowledgment and interest in our work! We sincerely appreciate your thorough and insightful comments on our work, and we will address each of your main concerns below:
>
> **Q1**: Conclusions in effectiveness verification of visual thought can be incoherence of the resulting context.
>
> **R1**:  Thank you for your constructive feedback. To address your concern, we compare direct prompting with GPT-4o to generate results (Direct) against explicitly instructing it to avoid generating any visual content (w/o VT prompting). As shown in Table 1, the results obtained by w/o VT prompting indicate a performance drop when visual information is will never be mentioned by model, even when the context is complete.
>
> It also aligns our major conclusion: Visual thoughts are the main visual information carrier.
>
> We will include more detailed discussions in the next version.
> > Table 1: Results on Direct prompting vs. w/o VT prompting.
> || CoMT-Selection | CoMT-Update | Math-Breakpoint |
> |:-:|:-:|:-:|:-:|
> | Direct | 13.00| 42.00| 84.00|
> | w/o VT prompting | 12.00| 32.00| 72.00 |
>
> **Q2**: The authors state that "removing visual thoughts leads to performance worse than using the query alone" maybe not correct.
>
> **R2**: Thank you for your meticulous and insightful comments. We sincerely believe there may have been a misunderstanding regarding our explanation. Actually, what we intended to convey is that in Figure 4(b), removing the visual thought module can, in some cases, lead to performance even lower than the Query-Only baseline. However, this does not imply that all instances of the “w/o Visual Thought” condition perform worse than the Query-Only baseline (as clarified in the first paragraph of Section 3).
>
> Moreover, this phenomenon is observed only in the Math subset, primarily because many Math questions contain sufficiently detailed textual descriptions that allow the model to perform reasoning without relying heavily on visual features. Removing visual thoughts in such cases disrupts the effective transmission of visual cues, which unexpectedly degrades performance. In contrast, for other datasets, the queries themselves carry minimal information (e.g., questions such as “How many people are in the image?”), and thus the Query-Only condition naturally performs worse than “w/o Visual Thought”.
>
> To further address your concerns, we have included additional experiments, as illustrated in Table 2. Similar trends are observed not only in GPT-4o but also in other models such as GPT-4o-mini and Gemini.
> >Table 2: Results on w/o visual thought v.s. Query-Only.
> || chess  | graph  |math|
> |-|:-:|:-:|:-:|
> |GPT-4o-mini w/o visual thought|6.00%|38.00% |9.00%|
> |GPT-4o-mini Query-Only|0.00%|3.00%|26.00%|
> |Gemini-2.0-flash w/o visual thought |4.00%| 24.00%|17.00%|
> |Gemini-2.0-flash Query-Only|1.00%|0.00%|29.00%|
>
> We will incorporate more detailed discussion on this issue in future versions of the paper to ensure clarity and alleviate any potential confusion.
>
> **Q3**: The problem with the baseline and suggestions of including the vanilla baseline without any CoT reasoning at all.
>
> **R3**: Thanks for your insightful feedback. The w/o VT of Section 4 and 'w/o visual thought' from Figure 4 represent the same meaning but not the same implementation as not including the visual thoughts in the rationale. Specifically, we implement methods in Section 4 by explicitly prompting the model and we implement methods in Figure 4 by removing the VT manually.
>
> In addition, we follow your suggestion to add the vanilla baseline without any CoT reasoning and the results are shown as:
> > Table 3: Results on w/o CoT.
> || MMVP | V\*Bench-Position | V\*Bench-attributes | M3CoT-physical | M3CoT-social | M3CoT-temporal | CoMT-deletion | CoMT-selection |CoMT-update|AVG|
> |-|-|-|-|-|-|-|-|-|-|-|
> | w/o VT | 74.33 | 53.95| 54.78| 88.89 | 76.86 | 79.67 | 26.50|19.50|37.00 |56.83 |
> | N-LANG |85.33| 57.89 | 63.48 | 88.89 | 78.93  | 83.74| 33.50| 25.50  |37.50|61.64|
> | S-LANG|84.33|63.16| 64.35| 90.00  | 78.51 | 82.93 | 29.50  | 18.00 |42.00|61.42|
> | E-IMG | 83.00 | 59.21 | 65.22  | 90.00 | 78.10 | 86.18 | 34.00| 28.50 |50.00 |63.80|
> | G-IMG| 78.00 | 59.21 | 59.13  | 92.22 | 78.93| 86.18 | 33.50 | 28.50  |46.50 |62.46|
> | w/o CoT | 77.67 | 57.89| 63.48| 87.78| 76.03 | 58.54  | 32.50 | 21.50 |41.50|57.43|
>
> We observe that the trends of removing Visual Thoughts (VT) and CoT are not consistent. As shown in Table 3, for benchmarks that require more reasoning steps, such as M3CoT, which involves extensive multi-step reasoning, the performance of the w/o CoT variant significantly degrades.
>
> Moreover, in most vision-reasoning scenarios, removing VT leads to even worse performance compared to removing CoT. This further substantiates our claim that visual thoughts are the core of MCoT. Rather than merely providing an additional reasoning process, VT plays a crucial role in conveying visual information throughout the reasoning pipeline.
>
> We will add more discussion in the next version.
>
> **Q4**: The conclusions drawn from the attention plots are not entirely convincing.
>
> **R4**: Thank you for your constructive feedback.
>
> **Attention Distribution Analysis**: To further support the conclusions drawn in Figure 7, we perform additional ***quantitative analyses*** to four models with varying layer counts: LLaVA-1.5-7B (32 layers), LLaVA-1.5-13B (40 layers), Qwen2.5-VL-3B (36 layers), and Qwen2.5-VL-7B (28 layers).
> 1. For each model, we extract internal attention to the original image using the NO-VIS method and compare it to attention directed at visual thoughts using the N-LANG, S-LANG, E-IMG, and G-IMG methods, as shown in Figure 7(b).
> 2. We then compute the percentage of layers where attention to visual thoughts exceed that to the input image.
> 3. This percentage is also calculated specifically for the latter 50% of layers (values shown in "()").
> > Table 4: The proportion of model layers in which the attention associated with visual thoughts surpasses that of the original input image. Values in "()" correspond to results calculated from the latter 50% of the model’s layers.
> ||Layer |N-LANG(%)|S-LANG(%) |E-IMG(%)|G-IMG(%)|
> |:-:|:-:|:-:|:-:|:-:|:-:|
> |LLaVA-1.5-7B|32|43.75 (12.50)|40.63 (6.25)| 96.88 (100.00) | 93.75 (100.00) |
> |LLaVA-1.5-13B|40|20.00 (0.00)|35.00 (5.00) | 95.00 (100.00) | 92.50 (100.00) |
> |Qwen2.5-VL-3B| 36 |97.22 (100.00) | 86.11 (100.00) | 72.22 (88.89) | 58.33 (100.00) |
> |Qwen2.5-VL-7B| 28|100.00 (100.00)| 92.86 (100.00) | 60.71 (100.00) | 60.71 (100.00) |
>
> **Results:**
> - Table 4 shows that in most layers across both LLaVA and Qwen models, attention favors visual thoughts over the original image, suggesting that reasoning relies more on visual thoughts.
> - Notably, in the latter half of the layers, attention to visual thoughts often dominates entirely, reaching 100% in some cases, indicating their superior capacity to convey visual information. These findings align with our overall conclusions.
>
> \* Under the N-LANG and S-LANG settings for LLaVA, attention to visual thoughts is relatively low, likely due to the model’s limited self-generated language capacity. The resulting captions and scene graphs are overly simplistic, failing to capture visual content adequately.
> > Table 5: The difference in attention score to visual thoughts relative to the original image across all layers.
> || Layer | N-LANG | S-LANG| E-IMG | G-IMG|
> |:-:|:-:|:-:|:-:|:-:|:-:|
> |LLaVA-1.5-7B| 32 | -0.0529 |-0.0472| 0.0156 | 0.0256|
> |LLaVA-1.5-13B |40| -0.0481| -0.0275| 0.0552 | 0.0637 |
> |Qwen2.5-VL-3B|36|0.1280| 0.0438| 0.0229 | 0.0388 |
> |Qwen2.5-VL-7B|28|0.1069| 0.0553  | 0.0189 | 0.0152|
>
> **Average Attention Score Anlaysis:** We compute the attention score differences induced by visual thoughts relative to the original image, averaged across all layers. The findings are:
> - As shown in Table 5, both LLaVA (7B->13B) and Qwen(7B->3B) exhibit increased attention gains with deeper model layers.
> - For Qwen2.5-VL, the 3B variant, despite its smaller parameter size, exhibits a greater attention increase than the 7B variant, likely due to its deeper architecture. This indicates that the influence of visual thoughts is more dependent on model design than parameter count.
>
> We will emphasize this section further and expand the discussion in the next version.
>
> **Q5**:  What metric is plotted on the y-axis of Figure 5? Why do all columns add up to 1? Please clarify in the caption or main text.
>
> **R5**: Thank you for your kind reminder. The y-axis in Figure 5 represents the normalized relative accuracy improvement of each dataset incorporating visual thoughts. The y-axis metric of $i$-th dataset can be mathematically represented as:
>
> $Metric_i^y = \frac{\delta acc_i}{\sum_{j=1}^{n} \delta acc_j},$
>
> where $\delta acc$ represents the relative accuracy improvement, $n$ represents the total number of datasets.
> We then calculate this metric for each type of visual thought, which produces the results in Figure 5 in our paper.
>
> We will add the details in the next version.
>
> **Q6**: Questions about the experiments on disruption of information flow.
>
> **R6**: Thank you for your valuable suggestion. In the information flow disruption experiment presented in Section 5.2, we adopt the methodology outlined in [1,2], in which the information exchange between designated token pairs is obstructed by assigning a value of $-\infty$ in selected transformer layers.
>
> We will provide more explanation and clarification in the next version.
>
> [1] Label words are anchors: An information flow perspective for understanding in-context learning.
>
> [2] Lifting the Veil on Visual Information Flow in MLLMs: Unlocking Pathways to Faster Inference.
>
> **Q7**: Typo Problems
>
> **R7**: Thanks for your valuable feedback. We will follow your suggestions to fix all typo problems in the next version.
>
> **Hope our clarification can address your concerns, and we sincerely hope that you can reconsider our work in light of these clarification.**

---

> > ### Author Response · Authors · 2025-08-02
> >
> > Thank you very much for your time and valuable feedback. Hope our clarifications can address your concerns, and we sincerely hope that you can reconsider our work in light of these clarifications. If you have any further comments, please do not hesitate to contact us. We greatly appreciate your selíess contributions to the community.

---

> > ### Comment · Reviewer_Z1x4 · 2025-08-03
> >
> > I would like to thank the Authors for the extensive and detailed rebuttal. Most of my concerns have been addressed, and I am happy to adjust my score accordingly.
> >
> > I believe readers could benefit from the clarifications provided in the rebuttal (for instance, those regarding the interpretation of Figure 4B, a concern shared also by Reviewer 79f4) and encourage the Authors to include them in the next version as they see fit.

---

> > > ### Author Response · Authors · 2025-08-03
> > > **Thanks for Your Time and Effort**
> > >
> > > We sincerely thank you for investing your time to review our response. Your insightful and valuable suggestions have significantly contributed to enhancing our work. We will follow your suggestions to enrich our work in the next version.
> > >
> > > Thank you once again for your valuable contributions.

---

### Official Review · Reviewer_NrVc · 2025-07-01

**Clarity:** 3
**Significance:** 2
**Originality:** 3
**Rating:** 4
**Confidence:** 3

**Summary:**

The authors present a taxonomy and analysis of multimodal chain-of-thought (MCoT) reasoning, categorizing thoughts into two main types: textual and visual, with four subtypes—natural language, structured language, image editing, and image generation thoughts. The study emphasizes the distinct roles of textual vs visual thoughts, supported by quantitative experiments on four datasets. The analysis includes computing correlation, flow, and attention scores to understand the impact of each thought type. Key findings include visual thoughts are crucial for enhancing MCoT reasoning, with their effectiveness depending on conciseness and clarity. Also, input image information is propagated to deeper transformer layers primarily through visual thoughts.

**Questions:**

1. The finding in Section 5.1 is interesting. I’m curious whether this phenomenon only appears in deep models. Furthermore, is visual thought more beneficial for deeper models? Could you test this with a quantitative experiment?

2. The differences between visual and textual thoughts in Flow and Disturbation might be related to the LLAVA training pattern. If you have access to models with different training patterns, it would be interesting to see whether these phenomena still hold.

3. In Figure 6, manually annotating relevance may limit the data scale. While human annotation is valuable and should be kept, you might also consider using auto methods (e.g., CLIP) to generate scores.

4. Beyond image similarity, I’d like to better understand the role of visual elements through representation analysis—e.g., by examining similarity between visual thought key/value pairs in attention.

**Ethical Concerns:**

["NO or VERY MINOR ethics concerns only"]

**Final Justification:**

This paper provides a detailed analysis of the mechanisms of visual thought and will help the community gain a better understanding of VLM inference. Although it does not propose a significant new method, I still find it valuable and above the bar.

**Limitations:**

no potential negative societal impact

**Paper Formatting Concerns:**

no formatting concerns

**Quality:**

3

**Strengths And Weaknesses:**

Strengths:

* Interesting topic with a clear categorization of multimodal CoT.
* Insightful analysis with good visualizations.
* Multiple perspectives and interesting findings.

Weaknesses:
* Some analyses could go deeper, would benefit from more extended results.

---

> ### Author Rebuttal · Authors · 2025-07-31
>
> Thanks for your acknowledgment and interest in our work! We sincerely appreciate your thorough and insightful comments on our work, and we will address each of your main concerns below:
>
> **Q1**: Is the finding in Section 5.1 only appearing in deep models? Is visual thought more beneficial for deeper models?
>
> **R1**: Thank you for your constructive feedback. To investigate how visual thoughts influence internal attention across models of different depths. We select four models with varying numbers of layers: LLaVA-1.5-7B (32 layers), LLaVA-1.5-13B (40 layers), Qwen2.5-VL-3B (36 layers), and Qwen2.5-VL-7B (28 layers).
>
> Specifically, we extract the model’s internal attention to the original input image during inference using the method without incorporating visual thoughts (no-vis), and extract the internal attention to visual thoughts using methods that incorporate visual thoughts (n-lang, s-lang, e-img, g-img), following the approach illustrated in Figure 7(b) of the paper. This allows us to compare the shifting trend of attention after integrating visual thoughts.
>
> We report the percentage of layers in which the attention score to visual thoughts exceeds that to the input image, as well as the percentage within the latter 50% of layers (shown in parentheses). The results are as follows:
> > Table 1: The percentage of layers in which the attention to visual thoughts exceeds that to the input original image. The number in parentheses indicates the result only discussed in the latter 50% layers of the model.
> || Layer |N-LANG(%)| S-LANG(%) | E-IMG(%)| G-IMG(%)|
> |:-:|:-:|:-:|:-:|:-:|:-:|
> | LLaVA-1.5-7B    | 32    | 43.75 (12.50)  | 40.63 (6.25)   | 96.88 (100.00) | 93.75 (100.00) |
> | LLaVA-1.5-13B   | 40    | 20.00 (0.00)   | 35.00 (5.00)   | 95.00 (100.00) | 92.50 (100.00) |
> | Qwen2.5-VL-3B   | 36    | 97.22 (100.00) | 86.11 (100.00) | 72.22 (88.89)  | 58.33 (100.00) |
> | Qwen2.5-VL-7B   | 28    | 100.00 (100.00)| 92.86 (100.00) | 60.71 (100.00) | 60.71 (100.00) |
>
> As shown in Table 1, **for both the LLaVA and Qwen models with various number of layers**, in majority of layers, the attention of visual thoughts exceeds that of the original image, **demonstrating that the majority attention are distributed to visual thoughts during the reasoning process**.
>
> Furthermore, the results in "()" indicate that in the latter 50% layers of the model, attention to visual thoughts almost entirely surpasses that to the original image, even 100% visual thought attention is higher than original image, suggesting that **visual thoughts can convey visual information more deeply than the original image itself**. These findings are consistent with our observations in the paper.
>
> \* In the n-lang and s-lang settings for LLaVA, the model shows relatively low attention to visual thoughts due to the self-generated weaker language capabilities, resulting in captions and scene graphs that are overly simplistic and fail to accurately convey visual information.
>
> We further compute the difference in attention score to visual thoughts relative to the original image, and average the results across all layers. The results are as follows:
> > Table 2: The difference in attention score to visual thoughts relative to the original image across all layers.
> || Layer | N-LANG | S-LANG| E-IMG | G-IMG|
> |:-:|:-:|:-:|:-:|:-:|:-:|
> | LLaVA-1.5-7B| 32 | -0.0529 | -0.0472  | 0.0156 | 0.0256|
> | LLaVA-1.5-13B | 40| -0.0481| -0.0275  | 0.0552 | 0.0637 |
> | Qwen2.5-VL-3B| 36| 0.1280| 0.0438| 0.0229 | 0.0388 |
> | Qwen2.5-VL-7B| 28 | 0.1069| 0.0553  | 0.0189 | 0.0152|
>
> As shown in Table 2, both LLaVA (7B->13B) and Qwen(7B->3B) exhibit increased attention gains with deeper model layers. In the case of Qwen2.5-VL, the 3B variant, **which has more layers, shows a more significant increase in attention** compared to the 7B variant. This suggests that the impact of visual thoughts depends more on the specific model architecture than on the parameter scale.
>
> **Q2**: The findings about information flow can be related to the LLAVA training pattern and it would be interesting to explore the different models.
>
> **R2**: Thank you for your valuable comment. We follow your suggestion to further investigate the distribution of information flow under the integration of the four types of visual thoughts using Qwen2-VL-2B.
> > Table 3: Information flow distribution of Visual Thought and Image using Qwen2-VL-2B.
> || N-LANG| S-LANG| E-IMG| G-IMG|
> |-|:-:|:-:|:-:|:-:|
> | original image to reasoning | 0.0028 | 0.0007 | 0.0022 | 0.0022 |
> | visual thought to reasoning | **0.0056** | **0.0010** | **0.0038** | **0.0038** |
>
> As shown in Table 3, we observe that visual thoughts can achieve larger information flow than the original image across all visual thought variants. Therefore, we can obtain the same conclusion as with the LLaVA models: *“The information flow from visual thoughts to the reasoning stages is considerably stronger than the direct flow from the image to reasoning.”*
>
> Furthermore, we observe that, in contrast to the behavior of LLaVA, Qwen2-VL achieves higher information flow distribution in the n-lang setting, owing to its stronger language generation capabilities.
>
> We will add more discussion in the next version.
>
> **Q3**: Auto methods (e.g., CLIP) can be beneficial for conclusions within Figure 6.
>
> **R3**: Thanks for your insightful feedback. To address the concerns regarding human evaluation bias and data scale limitations, we adopt your suggestion and use CLIP-based clarity scores to evaluate all visual thoughts.
>
> Specifically, we compute the classification probability between two categories: “This is a clear (annotated logical) image” and “This is not a clear (annotated logical) image” (for G-IMG, content in parentheses is removed). These probabilities serve as quantitative indicators for clarity score of visual thought.
> > Table 4: Correlation analysis between clip-based clarity scores and model performance on GPT-4o-mini.
> || Pearson Correlation| P |
> |-|:-:|:-|
> | S-LANG | 0.692| 0.06 |
> | N-LANG | 0.693| 0.04 |
> | G-IMG  | 0.731 | 0.04|
> | E-IMG  | 0.646| 0.06|
>
> As shown in Table 4, there is a clear correlation between clarity scores and model performance, indicating that high-quality visual thoughts are critical for achieving strong results. This aligns with our observations in paper.
>
> **Q4**: The role of visual thoughts through representation analysis.
>
> **R4**: Thank you for your constructive suggestions. Combining the results from Table 1 and Table 2, we observe that the internal attention mechanism reflects the effectiveness in conveying visual information of visual thoughts. For instance, in the case of LLaVA, which has relatively weaker language capabilities, the generated captions (N-LANG) or scene graphs (S-LANG) tend to be of lower quality and fail to effectively convey visual information, resulting in lower attention scores. This highlights the correlation between visual thoughts and the model’s internal representations.

---

> > ### Author Response · Authors · 2025-08-02
> >
> > Thank you very much for your time and valuable feedback. Hope our clarifications can address your concerns, and we sincerely hope that you can reconsider our work in light of these clarifications. If you have any further comments, please do not hesitate to contact us. We greatly appreciate your selíess contributions to the community.

---

> > ### Comment · Reviewer_NrVc · 2025-08-04
> >
> > Thank you for the new experiments and repsonse. I found the results interesting, especially the experiment on the number of layers which supports your claim in the paper. I will to keep my score as borderline accept.

---

### Official Review · Reviewer_7efu · 2025-07-03

**Clarity:** 1
**Significance:** 3
**Originality:** 2
**Rating:** 4
**Confidence:** 3

**Summary:**

This paper introduces the concept of visual thoughts as a unifying abstraction for intermediate reasoning steps in multimodal chain-of-thought (MCoT) systems. It categorizes four modalities of visual thought expressions (natural language, structured language, edited image, generative image) and empirically compares their effectiveness across a variety of benchmarks and large vision-language models (LVLMs). The authors further analyze attention patterns and information flow to argue that visual thoughts function as a “cache” that facilitates deeper reasoning.

This paper focuses on categorizing the idea of visual thoughts, and analysis of it, without clear technical contributions of new methods or datasets.  To improve the paper, show that explicitly designing models or prompts informed by the “visual thoughts” concept yields improvements over existing approaches.

**Questions:**

see above

**Ethical Concerns:**

["NO or VERY MINOR ethics concerns only"]

**Final Justification:**

after reading the rebuttal, other reviews and discussion, I've increased my score from BR to BA.  The rebuttal highlighted the contribution of the paper, and its transferrable insights.

**Limitations:**

yes

**Paper Formatting Concerns:**

no issues

**Quality:**

2

**Strengths And Weaknesses:**

# Strengths
* Provides a clear, structured categorization of intermediate visual reasoning mechanisms.
* Attention and information-flow studies provide some internal evidence supporting the “cache” hypothesis.
* Presents thorough experiments across several tasks and models, supported by ablation and diagnostic analyses.

# Weaknesses
* The core contribution — the concept of visual thoughts — is more an interpretive narrative than a substantive technical advance.
* The work primarily re-labels existing techniques (scene graphs, image editing, image generation, captions) rather than proposing new methods or benchmarks.

# Suggestions
* Benchmark directly against the most related prior works (e.g., Visual Sketchpad [15], Image-of-Thought [55], Scene Graph CoT) under comparable conditions.
* Add an experiment showing that explicitly designing prompts or models informed by the “visual thoughts” perspective achieves measurable improvements over existing baselines.
* Report stronger statistical analysis where possible: include confidence intervals, p-values, or more runs to strengthen claims about significance.

---

> ### Author Rebuttal · Authors · 2025-07-31
>
> Thanks for your acknowledgment and interest in our work! We sincerely appreciate your thorough and insightful comments on our work, and we will address each of your main concerns below:
>
> **Q1**: The concept of visual thoughts is more an interpretive narrative than a substantive technical advance, like new methods and benchmarks.
>
> **R1**: Thanks for your kind feedback. We sincerely believe there may be a misunderstanding. We sincerely think that the main contribution of our work is not to introduce new methods or benchmarks, but to present visual thoughts as a unifying perspective to better understand the mechanisms underlying MCoT (one of the most influential paradigms in the field). This contribution has also been acknowledged by other reviewers (Reviewer #NrVc, Reviewer #Z1x4, Reviewer #79f4).  From this viewpoint, our study yields several notable and thought-provoking findings:
> - First, we identify visual thoughts as the central functional component of MCoT. Regardless of their specific form, visual thoughts act as cache-like carriers of visual information during the reasoning process.
> - Second, we conduct a comparative analysis of various MCoT implementations and reveal that the key determinants of performance are the clarity and efficiency of these visual thoughts.
> - Third, we examine attention patterns and information flow to uncover the internal mechanisms by which visual thoughts serve as core intermediaries. Our findings suggest that they play a crucial role in sending more visual information to deeper model layers to produce the better reasoning trajectory of MCoT.
>
> We hope this study can serve as a  foundation for understanding how visual reasoning unfolds in large vision language models. We will **definitely** add more discussion in the next version to better address your technical concerns.
>
> **Q2**: Benchmark directly against the most related prior works (e.g., Visual Sketchpad [15], Image-of-Thought [55], Scene Graph CoT) under comparable conditions.
>
> **R2**: Thank you for your constructive feedback. Actually, the experiments shown in Figure 10 of our paper’s appendix were conducted directly on the Visual Sketchpad. In Table 1, our N-LANG setting is aligned with MM-CoT, the S-LANG setting corresponds to Scene Graph CoT, and the E-IMG setting is consistent with the Visual Sketchpad (note that the Image-of-Thought setting actually involves multi-turn visual reasoning).
>
> In addition, to ensure what you referred to as “under comparable conditions,” we constrained all methods to generate responses within two interaction rounds, thereby ensuring comparability. Moreover, our comparison is not only direct but also accompanied by a detailed analysis of their differences and the underlying reasons.
>
> Therefore, we have in fact conducted a comparison with the most relevant prior works under comparable conditions. We will revise the next version of the manuscript to clarify this point more explicitly.
>
> **Q3**: Add an experiment showing that explicitly designing prompts or models informed by the “visual thoughts” perspective achieves measurable improvements over existing baselines.
>
> **R3**:  Thanks for your insightful feedback. Inspired by self-consistency, we follow your suggestion to design a novel visual-thought-guided strategy that ensembles diverse visual thoughts reasoning paths, which are refered as diverse-VT. As shown in Table 1, we observe that Diverse-VT achieves the best performance.
> >Table 1: Result of Divserse-VT.
> || MMVP  | V\*Bench-Position | V\*Bench-attributes | M3CoT-physical | M3CoT-social |  M3CoT-temporal  | CoMT-deletion  | CoMT-selection  | CoMT-update | AVG |
> |:-|:-:|:-:|:-:|:-:|:-:|:-:|:-:|:-:|:-:|:-:|
> | w/o VT | 74.33 | 53.95 | 54.78   | 88.89   | 76.86  | 79.67 | 26.50 | 19.50 | 37.00 | 56.83 |
> | N-LANG (Maj@4)    | 85.00 | 57.89 | 63.48 | 88.89 | 78.93 | 83.74 | 33.50 | 25.50 | 37.50 | 61.60 |
> | S-LANG (Maj@4)    | 83.67 | 63.16  | 64.35  | 90.00 | 78.51 | 82.93 | 29.50 | 18.00 | 42.00 | 61.35 |
> | E-IMG (Maj@4)     | 82.33 | 59.21| 65.22   | 92.22  | 78.10  | 86.18 | 33.50 | 28.00 | 49.00 | 63.75|
> | G-IMG (Maj@4)     | 78.00 | 59.21 | 63.48    | 92.22    | 78.93   | 86.18 | 33.50 | 28.50 | 46.50 | 62.95|
> | Diverse-VT        | 85.00 | 63.16   | 65.22    | 92.22 | 78.93  | 86.18 | 34.00 | 28.50 | 50.00 | 64.80 |
>
> We will add more discussion in the next version.
>
> **Hope our clarifications can address your concerns, and we sincerely hope that you can reconsider our work in light of these clariìcations.**

---

> > ### Author Response · Authors · 2025-08-02
> >
> > Thank you very much for your time and valuable feedback. Hope our clarifications can address your concerns, and we sincerely hope that you can reconsider our work in light of these clarifications. If you have any further comments, please do not hesitate to contact us. We greatly appreciate your selíess contributions to the community.

---

### Note · Authors · 2025-08-13

We sincerely thank all reviewers for their insightful and valuable feedback.
- We are pleased that reviewers find our work to present an **interesting, well-motivated topic** (Reviewer #NrVc, Reviewer #Z1x4) that can guide future research in selecting or designing MCoT strategies (Reviewer #Z1x4).
- We are encouraged that our **structured and clear categorization** of MCoT reasoning (Reviewer #7efu,Reviewer #NrVc) and **systematic multi-task, multi-model evaluations** offering practical insights into MCoT’s suitability (Reviewer #Z1x4,Reviewer #79f4) are well appreciated.
- We are also glad that reviewers also value our **analysis of internal mechanisms**, which **provides evidence and interesting insights** into how visual information is conveyed to the reasoning process (Reviewer #7efu, Reviewer #NrVc, Reviewer #Z1x4), thus deepening the study (Reviewer #79f4).

In addition, to respond to thoughtful suggestions provided by reviewers, we add extensive experiments as follows:
- **Generality of internal mechanisms**: (1) For attention analysis, we additionally include  LLaVA and Qwen2.5-VL models, confirming our original findings and further revealing that VT can convey more information to deeper layers  rather than wider layer distributions. (2) For information flow, we add broader experiments on Qwen2-VL, which is also consistent with our LLaVA results in the original paper. These resolve the concern and are acknowledged by reviewers (Reviewer #NrVc, Reviewer #Z1x4, Reviewer #79f4).
- **Reproducibility of Figure 6**: To better improve reproducibility on the correlation analysis between VT-quality and performance, we re-evaluate using CLIPScore, obtaining results consistent with the manual evaluation results in the paper, which are recognized by reviewers (Reviewer #NrVc, Reviewer #79f4).
- **Sufficiency of Figure 4(b) discussion**: For the phenomenon where the performance of w/o vt on Math subset is lower than that of query-only, we conduct case analyses and extend the discussion with additional models and datasets to strengthen the discussion, which addresses the concern (Reviewer #Z1x4, Reviewer #79f4).
- **Other clarifications**: We supplement the w/o CoT baseline (Reviewer #Z1x4) and analyze noise from image-based visual thoughts (Reviewer #79f4), both acknowledged by reviewers.

We sincerely thank the reviewers for their constructive suggestions. We will carefully consider these points and incorporate the suggested discussions into the next version.

---

### Decision · Program_Chairs · 2025-09-17

**Decision:**

Accept (poster)

**Comment:**

This paper introduces the concept of Visual Thoughts as a unifying framework to interpret and analyze intermediate reasoning steps in Multimodal Chain-of-Thought (MCoT). All reviewers converged on a borderline accept rating. While initial concerns centered around limited methodological novelty and clarity, the detailed rebuttal and new results convinced reviewers of the paper’s utility.

This is not a “breakthrough” paper in the traditional NeurIPS sense, but it offers an important perspective that clarifies and systematizes an emerging area (MCoT). The empirical results and mechanistic insights could directly inform both academic and applied work in multimodal reasoning. Given this, the paper is recommended for poster acceptance.